# EDBench: Large-Scale Electron Density Data for Molecular Modeling

**Hongxin Xiang**[1,5], **Ke Li**[2], **Mingquan Liu**[1], **Zhixiang Cheng**[1], **Bin Yao**[3], **Wenjie Du**[4],
**Jun Xia**[5,6], **Li Zeng**[1], **Xin Jin**[7†], **Xiangxiang Zeng**[1†]

[1]College of Computer Science and Electronic Engineering, Hunan University
[2]College of Life Sciences, East China Normal University
[3]College of Materials Science and Engineering, Hunan University
[4]University of Science and Technology of China
[5]AIMS Lab, The Hong Kong University of Science and Technology (Guangzhou)
[6]The Hong Kong University of Science and Technology
[7]Eastern Institute of Technology

⌂ Main Page    ⛃ Data    ⬡ Code

## Abstract

Existing molecular machine learning force fields (MLFFs) generally focus on the learning of atoms, molecules, and simple quantum chemical properties (such as energy and force), but ignore the importance of electron density (ED) $\rho(r)$ in accurately understanding molecular force fields (MFFs). ED describes the probability of finding electrons at specific locations around atoms or molecules, which uniquely determines all ground state properties (such as energy, molecular structure, etc.) of interactive multi-particle systems according to the Hohenberg-Kohn theorem. However, the calculation of ED relies on the time-consuming first-principles density functional theory (DFT), which leads to the lack of large-scale ED data and limits its application in MLFFs. In this paper, we introduce EDBench, a large-scale, high-quality dataset of ED designed to advance learning-based research at the electronic scale. Built upon the PCQM4Mv2, EDBench provides accurate ED data, covering 3.3 million molecules. To comprehensively evaluate the ability of models to understand and utilize electronic information, we design a suite of ED-centric benchmark tasks spanning prediction, retrieval, and generation. Our evaluation of several state-of-the-art methods demonstrates that learning from EDBench is not only feasible but also achieves high accuracy. Moreover, we show that learning-based methods can efficiently calculate ED with comparable precision while significantly reducing the computational cost relative to traditional DFT calculations. All data and benchmarks from EDBench will be freely available, laying a robust foundation for ED-driven drug discovery and materials science.

## 1 Introduction

With the widespread adoption of deep learning in molecular dynamics (MD) simulations, machine learning force fields (MLFFs) have become efficient and promising computational tools, significantly advancing research in physics, chemistry, biology, and materials science [1, 2, 3]. State-of-the-art MLFFs methods typically employ geometric deep learning to model atomic interactions within molecules, a strategy that has proven to be effective [4]. These models are generally built upon

---

† Correspondence: `jinxin@eitech.edu.cn`, `xzeng@hnu.edu.cn`

39th Conference on Neural Information Processing Systems (NeurIPS 2025) Track on Datasets and Benchmarks.

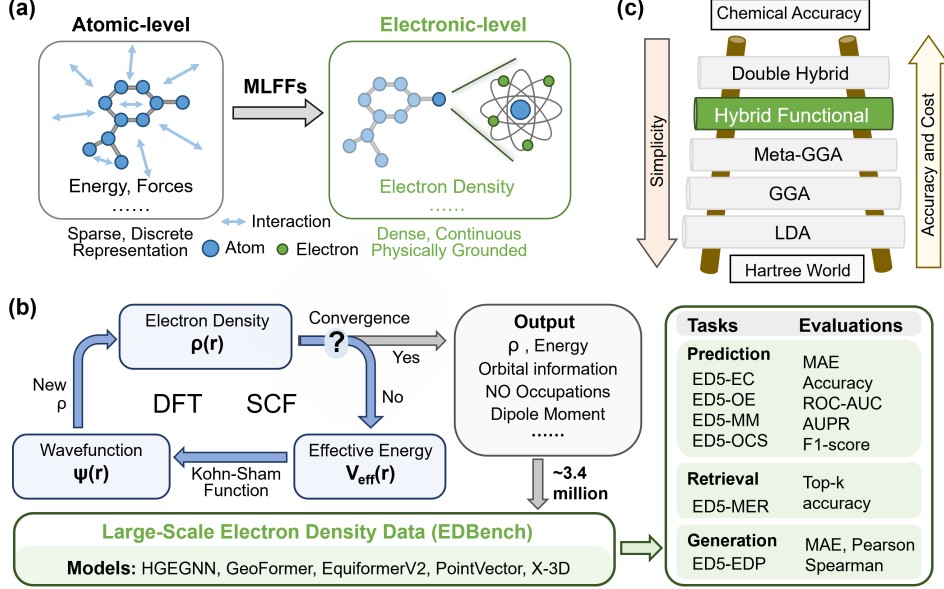

Figure 1: (a) Advancing MLFFs from atomic-level interactions—based on discrete atomistic representations—to electronic-level modeling using continuous ED, enabling richer and more physically grounded supervision; (b) Overview of the proposed EDBench dataset; (c) DFT method selection guided by Jacob's ladder to balance accuracy and computational cost.

many-body interactions at the atomic level, including one-body (atomic attributes such as types and coordinates [5]), two-body (interatomic distances [6, 2]), three-body (bond angles [7, 8]), four-body (torsions [9, 10, 11] and improper torsions [4]), and five-body interactions [12].

Although existing MLFFs have demonstrated great potential in modeling molecular force fields (MFFs), they primarily focus on capturing coarse-grained, atom-level many-body interactions [13, 4], while often overlooking the critical role of microscopic electron distribution in understanding molecular interactions [14, 15, 16]. It is well known that the spatial distribution of electrons directly influences the interactions between atoms within a molecule, providing the most direct and fundamental information for interpreting molecular force fields [17]. Electron density (ED), as a fundamental physical quantity in quantum mechanics that describes the spatial distribution of electrons, offers a more fine-grained and physically grounded representation of molecular systems according to Hohenberg–Kohn (HK) theorem [18]. Therefore, explicitly incorporating ED into the modeling process holds promise for bridging the gap between microscopic electronic behavior and macroscopic force fields, further improving both the accuracy and generalization of MLFFs. Therefore, as illustrated in Figure 1(a), the primary objective of this work is to advance current MLFFs beyond the atom-level learning paradigm toward electron-level modeling, enabling a more accurate and physically grounded description of molecular interactions. We provide a detailed description of the background in Appendix A.1 to enhance the reader's understanding.

However, advancing MLFFs toward an electron-level understanding faces two major challenges: (i) *the lack of large-scale, high-quality ED datasets*, which are essential for pretraining and could fundamentally reshape the paradigm of MLFFs modeling. (ii) *the absence of an ED-centric benchmark* to systematically explore the feasibility and effectiveness of ED-based modeling frameworks. Specifically, the acquisition of ED data can be categorized into two approaches: experimental methods (such as X-ray diffraction [19, 20], electron diffraction [21]) and theoretical calculation methods. Due to the reliance on expensive physical equipment, experimental methods inevitably limit data acquisition, making theoretical methods more popular. Theoretical calculations typically use density functional theory (DFT) [22, 23], the most common approach, to compute the ED of molecules. Although DFT does not depend on specialized observation equipment, its calculations are highly computationally intensive and time-consuming, making the acquisition of large-scale, high-quality ED datasets particularly difficult. In addition, the MLFFs community is still in the early stages of

learning effective representations from ED, which makes the development of an ED-based evaluation protocol particularly important for the rapid advancement of ED representation learning.

To address the two key challenges outlined above, we introduce EDBench, a large-scale and high-fidelity dataset of ED, as shown in Figure 1(b). Following Jacob's ladder [24], as shown in Figure 1(c), we adopt higher-rung hybrid functionals as the underlying DFT methods to ensure the quality of the EDBench dataset. Specifically, we use B3LYP function combined with 6-31G**/+G ** basis set and run over 205,000 core-hours on a high-performance computer to generate EDBench. EDBench comprises 3,359,472 drug-like molecules with corresponding ED distributions and a comprehensive set of quantum chemical properties, including energy components, orbital energies, and multipole moments, thus providing a solid foundation for systematically investigating the role of ED in molecular modeling. To enable rigorous benchmarking and model development, we further design an ED-centric benchmark suite covering three task categories: (i) quantum property prediction, including four core tasks—energy components prediction (ED5-EC), orbital energy estimation (ED5-OE), multipole moment regression (ED5-MM), and open-/closed-shell classification (ED5-OCS)—to evaluate how well ED alone can serve as a sufficient descriptor for inferring fundamental quantum properties; (ii) cross-modal retrieval between molecular structures and ED (ED5-MER), designed to probe the mutual consistency and representational alignment between structural and density spaces, which is critical for density-based force field construction and virtual screening; and (iii) ED prediction from molecular structures (ED5-EDP), aimed at approximating DFT-level density fields at significantly reduced computational cost, thereby enabling scalable quantum-aware modeling. Finally, we evaluate several state-of-the-art deep learning models on the proposed benchmark, offering the first large-scale assessment of ED understanding in data-driven systems.

## 2 Background and Related Works

### 2.1 Density Functional Theory (DFT)

The quantum mechanical description of many-electron systems is one of the core issues in modern physics and chemistry. Schrödinger equation [25] as the fundamental equation of quantum mechanics, is challenging to solve directly. Consequently, researchers introduced various wave function-based approximation methods to simplify the problem, such as, Born–Oppenheimer [26] and Hartree-Fock method [27]. Those methods scale with the number of electrons $n$ as $\mathcal{O}(n^4)$ or more, its computational cost remains prohibitive for large polyatomic molecules (More details about the development of quantum mechanics see Appendix A.2.1). In contrast, Density Functional Theory (DFT) is more suitable for complex systems due to its lower computational cost ($\mathcal{O}(n^3)$) and incorporation of electron correlation effects [28]. The core concept of DFT is to use electron density (ED) as the fundamental variable instead of the wave function. The Hohenberg-Kohn theorem is the cornerstone of DFT [18], which states that the external potential field and the ground-state energy can be completely determined by ED. Thus, by solving for the ED distribution $\rho(r)$ that achieves the lowest energy, the properties of the stable system can be confirmed. The ED $\rho(r)$ can be expressed as:

$$\rho(\mathbf{r}) = \rho_\alpha(\mathbf{r}) + \rho_\beta(\mathbf{r}) \tag{1}$$

where $\rho_\alpha(\mathbf{r})$ and $\rho_\beta(\mathbf{r})$ are the density of $\alpha$-spin electrons and $\beta$-spin electrons. This concept is concretely realized in the Kohn-Sham equations, which transforms the polyelectron system with interactions into single-electron system without interaction, and adds interactions among electrons to exchange-correlation potential [29]. The Kohn-Sham equations is shown as:

$$\left[ -\frac{1}{2} \nabla^2 + V_{\text{eff}}(r) \right] \psi_i(r) = \epsilon_i \psi_i(r) \tag{2}$$

where $\psi_i(r)$ and $\epsilon_i$ are, respectively, the wave function and energy of the $i$-th single-electron orbital, and $V_{\text{eff}}(r)$ is the effective single-electron potential energy (Details of $V_{\text{eff}}$ see Appendix A.2.2).

The solution of the Kohn-Sham equations is typically achieved through self-consistent field (SCF) iterations, as shown in the figure 1(b). Initially, a set of initial electron densities $\rho(\mathbf{r})$ is selected, and the effective potential $V_{\text{eff}}(\mathbf{r})$ is calculated based on this initial guess. The Kohn-Sham equations are then solved to obtain new single-electron orbital wave functions $\psi_i(\mathbf{r})$ and energies $\varepsilon_i$, which are used to update the ED $\rho(\mathbf{r})$. This process is repeated until convergence is achieved, yielding the ED $\rho(\mathbf{r})$ and simultaneously stabilizing the total energy $E$. The choice of the exchange-correlation functional

determines the accuracy of DFT, as shown in Figure 1(c). Common approximations include the Local Density Approximation (LDA) [30], the Generalized Gradient Approximation (GGA) [31], and hybrid functionals (B3LYP) [32]. In this paper, we used B3LYP functional combined with 6-31G**/+G** basis set, which achieves a balance between precision and efficiency(Details on basis set and functional see Appendix A.2.3).

## 2.2 Molecular Geometry Learning in Quantum Chemistry

Geometric Deep Learning (GDL) has become a dominant approach for modeling machine learning force fields (MLFFs), primarily focusing on atom-level information such as atomic attributes and interatomic interactions. Specifically, GDL models are built upon first-order atomic features, including atom types and 3D coordinates [5, 33]. To capture geometric relationships while preserving physical consistency, GDL methods incorporate symmetries such as rotational and translational invariance in 3D space [34, 35]. Consequently, a wide range of models have been developed with built-in invariance or equivariance to Euclidean group E(3) [5] or special Euclidean group SE(3) [36, 37], ensuring that predictions are physically meaningful. Given that atomic interactions—such as bonding, repulsion, and van der Waals forces—play a crucial role in molecular fields, modern GDL methods further incorporate second-order geometric features, including interatomic distances [38, 39], bond types [40], and spatial neighborhood structures [41]. To more precisely capture local structural features, some approaches even extend to higher-order geometric relations such as bond angles (three-body interactions) [7, 8] and torsional angles (four-body interactions) [9, 10, 11], thereby improving the expressiveness and accuracy of force field modeling. In contrast to prior works that focus primarily on atom-level representations, our proposed EDBench introduces a large-scale dataset of electronic density (ED), laying the foundation for extending molecular modeling from the atomic scale to the electronic scale. It also provides a new platform and evaluation benchmark for developing GDL methods tailored to electronic structure modeling.

## 2.3 Quantum Chemistry (QC) Datasets

To more efficiently predict quantum chemical properties and enhance applications using machine learning models, numerous QC datasets have been constructed to provide rich and reliable data support, as shown in Table 1. QM7 [42] and QM9 [43] are classic QC datasets, built based on GDB-13 [44] and GDB-17 [45], respectively. QM7-X [46] covers up to 42 physicochemical properties. PubQChemQC [47] offers 85 million ground-state molecular structures and HOMO-LUMO gaps. Additionally, MD17 [48], MD22 [49] and WS22 [50] offer force data from molecular dynamics trajectories. Based on QM9 dataset, QH9 [51] focuses on the Hamiltonian matrix, and MultiXC-QM9 [52] includes information on chemical reactions. Compared with QC datasets that provide energy [53] and force, the proportion of existing ED datasets is relatively small. Many datasets providing ED $\rho$ are concentrated in the field of material. For instance, MP [54] provides approximately 122K PBE-accuracy ED datas, and ECD [55] contains 140K PBE-accuracy entries and 7K entries with HSE-accuracy. This ED information provides CHGCAR files. Additionally, for drug-like molecules, QMugs [56] and $\nabla^2$DFT [57] provide density matrices for 665K and 1.9M molecules, respectively, but do not directly provide ED. Moreover, three ED datasets generated by VASP were used for the equivariant DeepDFT model [58], including QM9-VASP, but they have few molecules. As atomic-level MLFFs are fully researched, such methods gradually face bottlenecks, highlighting the increasing need for large-scale molecular datasets that incorporate ED and comprehensive benchmarks to facilitate subsequent research. In this study, we constructed a dataset named EDBench, which contains nearly 3.4 million molecules, including ED (CUBE files) and common quantum chemical properties. As shown in Table 1, compared with other datasets, EDBench provides the largest known ED benchmark dataset, characterized by its large scale and comprehensive content. We also summarize physical and chemical properties of different quantum datasets in Appendix A.3.

## 3 Dataset, Tasks, Methods, Evaluations

### 3.1 Dataset

**Construction of EDBench dataset.** We performed large-scale density functional theory (DFT) calculations on 3,359,472 molecules from the PCQM4Mv2 dataset [59] using Psi4 1.7 [60, 61] (We describe the reasons about why PubCheMQC [59] was not used in Appendix A.4). We employed

Table 1: Comparison of various databases in quantum chemistry.

| Category | Ours | Classical quantum chemistry databases and extensions | | | | | Molecular dynamics | | Pharmaceutical | | Material | | |
|---|---|---|---|---|---|---|---|---|---|---|---|---|---|
| Datasets | EDBench | QM7-X | QM9 | QH9 | QM9-VASP | PubChem QC | WS22 | MD17 | QMugs | $\nabla^2$DFT | QMMD | ECD | MP |
| Source | PCQM4Mv2 | GDB-13 | GDB-17 | QM9 | QM9 | PubChem | - | - | CHEMBL | MOSES | ICSD | Magten | ICSD |
| Molecules | 3,359,472 | 7K | 134K | 130K | 134K | 85M | 10 | 8 | 665K | 1.9M | 1.2M | 140K | 577K |
| Element Count | 22 | 6 | 5 | 5 | 5 | 12 | 4 | 4 | 11 | 8 | - | - | - |
| Calc Method (Basis Set/ XC-Function) | B3LYP, 6-31G**/+G** | PBE0+ MBD | B3LYP, 6-31G (2df,P) +G4MP2 | B3LYP, Def2-SVP | PBE | B3LYP/6-31G* //PM6 | PBE0/ 6-31G* | PBE+ vdW-TS | GFN2-x TB+$\omega$B97 X-D/def 2-SVP | $\Omega$B97X-D/Def2-SVP | PAW-PBE | PBE/HSE 06,GGA+ U/Monkho rst-Pack | PBE,GGA /GGA+U, SCAN/R2 SCAN |
| Electron density $\rho$ | ✓-CUBE | × | × | × | ✓-CHG CAR | × | × | × | × | × | × | ✓-CHG CAR | ✓(122K)-CHGCAR |
| Total Energy | ✓ | ✓ | × | × | × | ✓ | × | ✓ | ✓ | ✓ | ✓ | × | ✓ |
| NO Occupation | ✓ | × | × | × | × | × | × | × | × | × | × | × | × |
| Canonical Orbit | ✓ | ✓ | ✓ | × | × | ✓ | ✓ | × | ✓ | ✓ | × | × | × |
| Dipole Moment | ✓ | ✓ | ✓ | × | × | ✓ | ✓ | × | ✓ | ✓ | × | × | × |
| E-Structure | ✓ | × | × | × | ✓ | ✓ | ✓ | × | ✓ | ✓ | ✓ | ✓ | ✓ |
| Software/Tool | Psi4 | FHI-aims | Gaussian | PySCF | VASP | GAMESS | Gaussian | ORCA/ FHI-aims | Psi4 | Psi4 | VASP | VASP | VASP |

the widely used B3LYP hybrid functional due to its strong empirical performance across diverse molecular systems. The choice of reference wavefunction was determined by the spin multiplicity, which was computed from the number of unpaired electrons according to Hund's rule. Specifically, we used a restricted Hartree-Fock (RHF) reference for closed-shell systems (multiplicity = 1), and an unrestricted Hartree-Fock (UHF) reference for open-shell systems (multiplicity > 1) to allow for independent optimization of $\alpha$ and $\beta$ spin orbitals. Basis sets were assigned based on elemental composition. We used 6-31G** for molecules without sulfur, while for sulfur-containing molecules, we used 6-31+G** to incorporate diffuse functions that better capture the more delocalized and polarizable electron distributions of heavier atoms like sulfur. After achieving self-consistent field (SCF) convergence, we generated cube files containing electron density (ED) data from Equation 1 with a grid spacing of 0.4 Bohr, a padding of 4.0 Bohr, and a density fraction threshold of 0.85 to define the isosurface region. Examples of the ED visualization are shown in Appendix A.5. All computations were carried out on a high-performance server equipped with 8 Intel(R) Xeon(R) Platinum 8270 CPUs, each with 26 physical cores and 2 threads per core, yielding a total of 416 logical cores. The total computational cost exceeded 205,000 core-hours, equivalent to approximately 23.4 years of single-core compute time (See Appendix A.6 for details). Regarding data quality and reliability, refer to Appendix A.7. In addition, we verify the fidelity of different functionals to the ED in Appendix A.8 and Ti- and Zn-containing molecules in Appendix A.9.

**Statistical information.** Figure 2(a) shows the distribution of molecular lengths in EDBench, counting only heavy atoms (excluding hydrogens), with most molecules containing no more than 20. We explain in the Appendix A.10 why the maximum number of molecules is not more than 20. Figure 2(b) presents the distribution of heavy atom types, covering 21 distinct elements. Figures 2(c) and 2(d) show the distributions of ED lengths (the number of ED points) and the molecular mean ED values, respectively. The average ED length exceeds the average atomic length by more than four orders of magnitude.

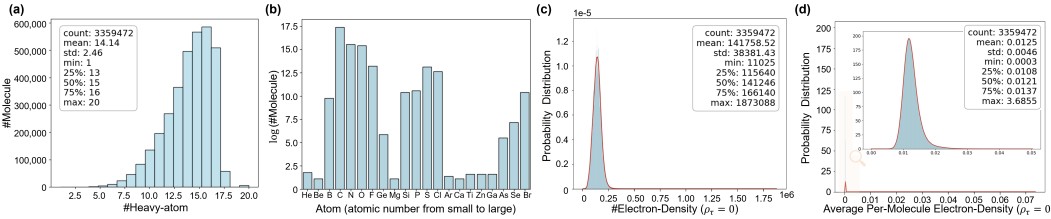

Figure 2: (a) Distribution of molecular lengths (heavy atoms only). (b) Distribution of heavy atom counts. (c) Distribution of ED lengths. (d) Distribution of per-molecule mean ED values.

## 3.2 Tasks

To comprehensively evaluate the capacity of the model to understand ED data, we define a suite of tasks based on both molecular structures (MS) and ED, focusing on three fundamental capabilities: prediction of quantum property, retrieval between MS and ED, and generation of ED based on MS (for the significance of these three types of tasks, see Appendix A.11). These tasks are constructed and conditionally sampled from the EDBench. To facilitate the development of ED-oriented machine learning methods within the community, we set the dataset size to a moderate scale of up to $n^{max} =$

$50,000$ molecules, with remaining data available for future research on pre-training strategies. We use scaffold split to divide the dataset into 80% training set, 10% validation set and 10% test set, which is an out-of-distribution split setting and is widely used to evaluate the generalization ability of the model [62, 63]. We summarize the statistics of the designed datasets in Table 2 (distribution details in Appendix A.12). We next explain the construction details of these tasks.

Table 2: Statistical information of designed 6 benchmarks with a scaffold split. ED5-EC, ED5-OE, ED5-MM, ED5-OCS, ED5-MER and ED5-EDP represent "Energy Components prediction", "Orbital Energy estimation", "Multipole Moment regression", "Open-/Closed-Shell classification", "MS and ED retrieval" and "ED Prediction", respectively.

| Datasets | #Mol | #Train/#Valid/#Test | #Task | Task type | Task desc |
|---|---|---|---|---|---|
| ED5-EC | 47,986 | 38,388/4,799/4,799 | 6 | Regression | 6 energy components (DF-RKS Final Energy [E1], Nuclear Repulsion Energy [E2], One-Electron Energy [E3], Two-Electron Energy [E4], DFT Exchange-Correlation Energy [E5], Total Energy [E6]) |
| ED5-OE | 43,510 | 34,808/4,351/4,351 | 7 | Regression | 7 orbital energies (HOMO-2, HOMO-1, HOMO-0, LUMO+0, LUMO+1, LUMO+2, LUMO+3) |
| ED5-MM | 49,917 | 39,933/4,992/4,992 | 4 | Regression | 4 multipole moment (3 Dipoles {X, Y, Z}, Magnitude) |
| ED5-OCS | 50,000 | 40,000/5,000/5,000 | 1 | Classification | open-/closed-shell classification |
| ED5-MER | 50,000 | 40,000/5,000/5,000 | 2 | Retrieval | cross-modal retrieval between molecular structures and ED |
| ED5-EDP | 50,000 | 40,000/5,000/5,000 | 1 | Generation | ED prediction from molecular structures |

**Prediction of quantum property.** To construct four task-specific datasets—ED5-EC (energy components), ED5-OE (orbital energies), ED5-MM (multipole moments), and ED5-OCS (open/closed-shell)—we design a structure- and label-balanced sampling strategy based on the full EDBench dataset ($n$ molecules). We first extract 2D ECFP4 fingerprints ($fp^{2D} \in \mathbb{R}^{n \times 2048}$) and 3D USR descriptors ($fp^{3D} \in \mathbb{R}^{n \times 12}$) for each molecule, concatenate them, and apply $k$-means clustering ($k = 100$) to obtain structure clusters $C^s$. For the multi-dimensional labels $y^{EP}$ (6D), $y^{GE}$ (7D), and $y^{MMR}$ (4D), we similarly apply $k$-means ($k = 100$) to produce clusters $C^{EC}$, $C^{OE}$, and $C^{MM}$, respectively; for $y^{OCS}$ (binary), we use the original label. We then form sampling groups as $(C^s, C^{EC})$, $(C^s, C^{OE})$, $(C^s, C^{MM})$, and $(C^s, y^{OCS})$, and uniformly sample $m = \max(n^{max}//n^{group}, 1)$ molecules from each group to construct the final datasets, ensuring diversity in both structure and property space.

**Retrieval between MS and ED.** Retrieval between MS and ED is a fundamental task. Retrieving molecular structures from ED (ED $\rightsquigarrow$ MS) enables electron-level virtual screening, while retrieving ED from structures (MS $\rightsquigarrow$ ED) supports electron-aware models—facilitating molecular representation learning, inverse design, and quantum-informed modeling. To construct the ED5-MER dataset for bidirectional retrieval between MS and ED, we group all molecules in EDBench by structure cluster $C^s$ and uniformly sample $m$ anchor (MS and ED) from each group. For each anchor, we sample $n^{neg} = 10$ negative samples: half from the same cluster (easy negatives) and half from different clusters (hard negatives). The final task involves identifying the correct ED (or MS) from a set of 11 candidates given an anchor MS (or ED).

**Generation of ED based on MS.** Generating ED from MS (MS $\rightarrow$ ED) is a highly valuable task, as it can significantly reduce the computational cost associated with DFT-based ED calculations. Since ED is inherently dependent on both molecular connectivity and 3D geometry, we ensure diversity in both structure and density by grouping molecules via $C^s$ and uniformly sampling $m$ MS-ED pairs from each group. In this task, the model is given an MS as input and is required to predict its ED.

### 3.3 Methods

To assess the model's understanding of electron density (ED), we design tailored learning paradigms for each task type (prediction, retrieval, and generation). For clarity, we formalize the molecular structure (MS) with $n$ atoms as $\mathcal{G} = (\mathcal{V}, \mathcal{Z}^{\mathcal{G}})$, where $\mathcal{V} = \{v_1, v_2, ..., v_n\} \in \mathbb{R}^{n \times 1}$ and $\mathcal{Z}^{\mathcal{G}} = \{z_1^{\mathcal{G}}, z_2^{\mathcal{G}}, ..., z_n^{\mathcal{G}}\} \in \mathbb{R}^{n \times 3}$ denote atomic types and their corresponding 3D coordinates, respectively. The ED data with $m$ points is denoted as $\mathcal{P} = (\mathcal{Z}^{\mathcal{P}}, \mathcal{D})$, where $\mathcal{Z}^{\mathcal{P}} = \{z_1^{\mathcal{P}}, z_2^{\mathcal{P}}, ..., z_m^{\mathcal{P}}\} \in \mathbb{R}^{m \times 3}$ represents the ED coordinates and the corresponding density values $\mathcal{D} = \{d_1, d_2, ..., d_m\} \in \mathbb{R}^{m \times 1}$. We denote the MS encoder and ED encoder as $\text{Enc}_{\mathcal{G}}$ and $\text{Enc}_{\mathcal{P}}$, respectively, to extract latent representations from MS and ED.

**For prediction tasks**, we introduce an additional task-specific prediction head $\texttt{Enc}_t$, whose output dimension matches the number of target labels for each task. The learning paradigm is defined as follows: the ED encoder $\texttt{Enc}_{\mathcal{P}}$ first extracts features from $\mathcal{P}$, which are then passed through $\texttt{Enc}_t$ to generate task-specific predictions $\hat{y}$. This process can be formalized as:

$$\hat{y}^{\bullet} = \texttt{Enc}_t^{\bullet}(\texttt{Enc}_{\mathcal{P}}(\mathcal{P})) \tag{3}$$

where $\bullet$ denotes a specific task, such as EC, OE, MM, or OCS. Accordingly, on the ED5-EC, ED5-OE, ED5-MM, and ED5-OCS datasets, we compute the loss between $\hat{y}^{\bullet}$ and the corresponding ground truth $y^{\bullet}$ to optimize the model. Specifically, cross-entropy loss is used for classification tasks, while L2 loss is applied for regression tasks.

**For retrieval tasks**, we utilize $\texttt{Enc}_{\mathcal{G}}$ and $\texttt{Enc}_{\mathcal{P}}$ to extract latent representations $h_{\mathcal{G}}$ and $h_{\mathcal{P}}$ from the MS $\mathcal{G}$ and ED $\mathcal{P}$, respectively, which can be formalized as:

$$h_{\mathcal{G}} = \texttt{Enc}_{\mathcal{G}}(\mathcal{G}), \quad h_{\mathcal{P}} = \texttt{Enc}_{\mathcal{P}}(\mathcal{P}) \tag{4}$$

The models are trained with the InfoNCE loss [64], which pulls matched pairs closer in the embedding space while pushing apart mismatched ones. Formally, given a batch of $n$ paired samples $\{(\mathcal{G}_i, \mathcal{P}_i)\}_{i=1}^n$, the loss for a positive pair $(\mathcal{G}_i, \mathcal{P}_i)$ is defined as:

$$\mathcal{L}_{\text{ret}} = -\log \frac{\exp(\text{sim}(h_{\mathcal{G}_i}, h_{\mathcal{P}_i})/\tau)}{\sum_{j=1}^n \exp(\text{sim}(h_{\mathcal{G}_i}, h_{\mathcal{P}_j})/\tau)} \tag{5}$$

where $\text{sim}(\cdot, \cdot)$ denotes a similarity function (e.g., cosine similarity), and $\tau = 0.07$ is a temperature.

**For the generation task**, we construct a heterogeneous graph [65], defined as:

$$\mathcal{HG} = \{(\mathcal{V}, \mathcal{Z}^{\mathcal{G}}), (\mathcal{Z}^{\mathcal{P}}, \mathcal{D}), \mathcal{E}\} \tag{6}$$

where $\mathcal{HG}$ contains two types of nodes: atoms and electrons. To construct the edge set $\mathcal{E}$, we perform a $k$-nearest neighbor search ($k = 9$) for each node, retrieving the $k$ closest nodes of the same type and $k$ of the opposite type, which results in 18 edges per node, forming atom–atom, atom–electron, and electron–electron connections. Since the goal is to predict ED from MS, we mask all ED values to obtain the masked graph $\hat{\mathcal{HG}}$. We extend Equivariant Graph Neural Network (EGNN) [5], called HGEGNN, to support heterogeneous graph. In HGEGNN, we treat electrons as special atoms and apply the same EGNN operations as used for regular atoms. We then input $\hat{\mathcal{HG}}$ into an HGEGNN to extract node representations $h^{\mathcal{HG}}$, which are split into atomic features $h_{\mathcal{G}}^{\mathcal{HG}}$ and electronic features $h_{\mathcal{P}}^{\mathcal{HG}}$. Finally, we apply a prediction head $\texttt{Enc}_t^{\text{EDP}}$ to the electronic features to generate the masked density values:

$$h^{\mathcal{HG}} = \text{HGEGNN}(\hat{\mathcal{HG}}), \quad \hat{\mathcal{D}} = \texttt{Enc}_t^{\text{EDP}}(h_{\mathcal{P}}^{\mathcal{HG}}) \tag{7}$$

where $\hat{\mathcal{D}} \in \mathbb{R}^{n_{\mathcal{P}} \times 1}$ is the predicted ED. We minimize the discrepancy between $\hat{\mathcal{D}}$ and the ground-truth $\mathcal{D}$ by the following L2 loss:

$$\mathcal{L}_{\text{gen}} = \|\hat{\mathcal{D}} - \mathcal{D}\|_p, \quad p = 2 \tag{8}$$

### 3.4 Evaluations

**In the prediction tasks**, the predicted labels are obtained via Equation 3. Specifically, in ED5-EC, ED5-OE, and ED5-MM, we evaluate the prediction performance using MAE between the predicted and ground-truth values, i.e., $(\hat{y}_{\mathcal{P}}^{EC}, y_{\mathcal{P}}^{EC})$, $(\hat{y}_{\mathcal{P}}^{OE}, y_{\mathcal{P}}^{OE})$, and $(\hat{y}_{\mathcal{P}}^{MM}, y_{\mathcal{P}}^{MM})$. For ED5-OCS, we assess classification performance using accuracy, ROC-AUC, AUPR, and F1-score between the predicted logits $\hat{y}_{\mathcal{P}}^{OCS}$ and ground-truth labels $y_{\mathcal{P}}^{OCS}$. **In the retrieval task**, we evaluate the quality of the latent features $h_{\mathcal{G}}$ and $h_{\mathcal{P}}$ extracted via Equation 4. Specifically, in ED5-MER, given a molecular feature $h_{\mathcal{G}_i}$ as the anchor, we retrieve from a set of ED features by computing cosine similarities and ranking the results; Top-$k$ accuracy ($k = 1, 3, 5$) is used as the evaluation metric, where a hit is counted if the correct match appears in the top $k$ results. Similarly, we perform retrieval in the opposite direction using $h_{\mathcal{P}_i}$ as the anchor and $h_{\mathcal{G}}$ as the candidate set. **In the generation task**, the predicted ED $\hat{\mathcal{D}}$ is obtained via Equation 7. We evaluate the generation performance using MAE, Pearson correlation coefficient and Spearman's rank correlation coefficient between predicted ED $\hat{\mathcal{D}}$ and the ground-truth $\mathcal{D}$ in ED5-EDP.

## 4 Experiment and discussion

**Baseline.** For comprehensiveness of the evaluation, we evaluate both molecular structure-based and electron density-based methods. Specifically, we selected several state-of-the-art baselines for evaluation on the proposed benchmark: (i) two geometric models based on molecular structure (MS): GeoFormer [66] and EquiformerV2 [67]; (ii) two point cloud models based on electron density (ED): PointVector [68] and X-3D [69]. GeoFormer and EquiformerV2 are Transformer-based architectures that use Interatomic Positional Encoding (IPE) and higher-degree tensors, respectively, to learn the interaction relationships between atoms. Unlike GeoFormer and EquiformerV2, which are specifically designed for molecules, PointVector and X-3D are the latest methods that focus on real-world point clouds. They are MLP(Multi-layer Perceptron)-based and explict structure-based architectures, respectively, offering excellent computational efficiency to handle large-scale point clouds. A comparison of the computational efficiency across different models is provided in Appendix A.13.

**Setup.** The codes of all baselines are available from their GitHub repositories and we reproduce them on our benchmarks. We use the same experimental settings as these baselines. All datasets are split using a scaffold split [62] based on the out-of-division (OOD) scenario, which enables evaluating the generalization of the model. We repeat the experiments three times with different random seeds and report the means and standard variances on the test set. The test set results are selected according to the best validation set performance. Due to the excessive length of the ED vectors (Figure 2(c)), we introduce a threshold $\rho_\tau$ to filter out electrons in regions with negligible density (all ED values below $\rho_\tau$ are discarded). All models were trained using either NVIDIA A100 (80GB PCIe) or GeForce RTX 3090 (24GB) GPUs, depending on their memory requirements.

### 4.1 Performance on prediction tasks

Table 3: The MAE performance on 6 energies from the ED5-EC dataset with $\rho_\tau = 0.05$.

|  | E1 | E2 | E3 | E4 | E5 | E6 |
|---|---|---|---|---|---|---|
| PointVector | 243.49±74.72 | 325.65±160.17 | 858.77±496.74 | 389.24±217.51 | 17.54±10.85 | 243.49±74.73 |
| X-3D | **190.77±1.98** | **109.21±2.82** | **369.88±1.34** | **150.05±0.27** | **8.13±0.51** | **190.77±1.98** |

Table 4: The performance of MAE×100 on 7 orbital energies of the ED5-OE with $\rho_\tau = 0.05$.

|  | HOMO-2 | HOMO-1 | HOMO-0 | LUMO+0 | LUMO+1 | LUMO+2 | LUMO+3 |
|---|---|---|---|---|---|---|---|
| PointVector | **1.73±0.01** | **1.68±0.01** | **1.92±0.01** | **3.08±0.05** | **2.86±0.05** | **3.05±0.02** | **3.01±0.02** |
| X-3D | 1.75±0.02 | 1.72±0.02 | 1.98±0.00 | 3.21±0.01 | 3.02±0.02 | 3.25±0.04 | 3.20±0.03 |

Table 5: The MAE performance on multipole moments from the ED5-MM dataset with $\rho_\tau = 0.05$.

|  | Dipole X | Dipole Y | Dipole Z | Magnitude |
|---|---|---|---|---|
| PointVector | 0.9123±0.0203 | 0.9605±0.0053 | 0.754±0.0068 | 0.7397±0.0467 |
| X-3D | **0.8818±0.0010** | **0.9427±0.0008** | **0.7416±.0023** | **0.6820±0.0005** |

Tables 3, 4, 5, and 6 report the performance of recent models on the ED5-EC, ED5-OE, ED5-MM, and ED5-OCS datasets, respectively. We observe that X-3D consistently outperforms PointVector, achieving the best results on ED5-EC (Ta-

Table 6: The performance (%) of open/closed-shell prediction on the ED5-OCS dataset with $\rho_\tau = 0.05$.

|  | Accuracy | ROC-AUC | AUPR | F1-Score |
|---|---|---|---|---|
| PointVector | 55.57±2.14 | 55.97±5.17 | 57.62±3.91 | **66.96±2.08** |
| X-3D | **57.65±0.18** | **60.48±0.38** | **61.54±0.31** | 61.41±1.02 |

ble 3), ED5-MM (Table 5), and ED5-OCS (Table 6). Notably, both X-3D and PointVector exhibit significantly stronger performance on orbital energy prediction (Table 4) than on energy component prediction (Table 3). This is likely due to the stronger locality of orbital energies, which are more directly linked to local ED patterns, allowing models to extract relevant features more effectively. In contrast, predicting energy components requires integrating over the entire ED, demanding the learning of more complex global interactions. In addition, see Appendix A.14 for the results of a training with full corpus (about 2.67 million molecules) on ED5-OE task and see Appendix A.15

for extension effectiveness study to periodic systems (materials). Overall, these results validate the effectiveness of using ED as a model input and demonstrate its utility in capturing physically meaningful patterns.

## 4.2 Performance on retrieval tasks

We employ {GeoFormer (G), EquiformerV2 (E)} and {PointVector (P), X-3D (X)} as the MS encoder $Enc_{\mathcal{G}}$ and ED encoder $Enc_{\mathcal{P}}$, respectively, in Equation 3. These components are paired to form four combinations: G+P, G+X, E+P, and E+X. Their retrieval performance is evaluated in Figure 3. The results show that combinations involving E (i.e., E+P and E+X) consistently outperform those involving G, highlighting the importance of selecting an appropriate encoder for retrieval tasks. Overall, the strong performance of E+P and E+X demonstrates their potential for ED-based virtual screening and MS-based electronic-level molecular understanding. For more detailed retrieval

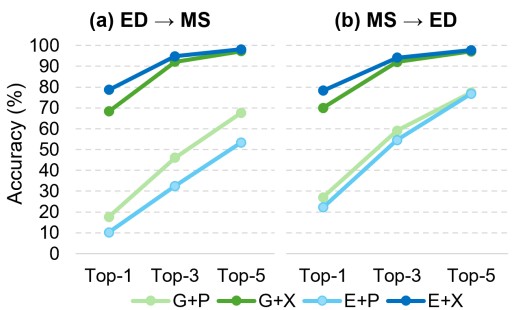

Figure 3: The retrieval performance on ED5-MER.

results, please refer to Appendix A.16 and see Appendix A.17 for the sensitivity analysis of negative samples.

## 4.3 Performance on generation task

Table 7 presents the results of HGEGNN on the ED prediction task under different ED thresholds $\rho_\tau$. We observe that, given the molecular structure (MS), the model can accurately predict ED values, achieving low MAE and high Pearson and Spearman correlations. This indicates that the deep learning method can signifi-

Table 7: The performance of HGEGNN on ED generation of ED5-EDP dataset. The unit of Time is second/mol.

|  | $\rho_\tau$ | MAE | Pearson (%) | Spearman (%) | Time |
|---|---|---|---|---|---|
|  | 0.1 | 0.3362±0.2900 | 81.0±8.1 | 56.4±13.7 | 0.024 |
| HGEGNN | 0.15 | 0.0463±0.0157 | 98.0±6.3 | 87.0±2.7 | 0.015 |
|  | 0.2 | 0.0448±0.0133 | 99.2±0.8 | 91.0±9.1 | 0.013 |
| DFT | - | - | - | - | 245.8 |

cantly accelerate the generation of ED while reducing the computational cost associated with DFT calculations. Notably, the model performance improves with increasing $\rho_\tau$, indicating it effectively captures high-ED regions. This aligns with chemical intuition, as high-density regions often correspond to chemically significant areas such as atomic cores and bonding regions, where the spatial patterns are more structured and consistent across molecules, making them easier for the model to learn. We show the element-wise and size-resolved error statistics in Appendix A.18.

## 4.4 Quality analysis of ED outputs from the generation task

To assess the quality of the ED data generated by HGEGNN, we employ models trained with three different random seeds, as described in Section 4.3, to generate ED5-EC data with a density threshold of $\rho_\tau = 0.2$, denoted as G#1, G#2, and G#3. Figure 4 compares the average MAE performance of different data sources using the PointVector as baseline, where red values denote relative improvements compared to DFT-based data source. We observe that G#1, G#2, and G#3 consistently outperform the DFT-based data, indicating that HGEGNN generates high-quality ED. These demonstrate the potential of using predicted ED directly to enhance the model's understanding of MFFs. Notaby, the superior downstream performance of model-generated EDs does not necessarily indicate higher physical fidelity; rather, it may result from their smoother, more learnable patterns that align better with the inductive biases and

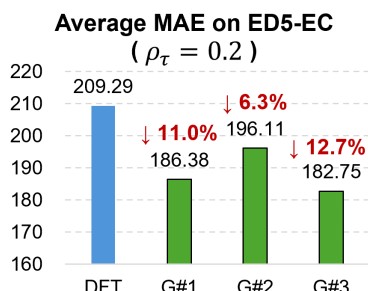

Figure 4: The average MAE of PointVector on ED5-EC generated by DFT and G#{1,2,3}.

optimization dynamics of downstream models than DFT-based EDs. More detailed results refer to Appendix A.19.

### 4.5 Ablation study on thresholds and sampling points

Due to the substantial number of ED points and their direct influence on computational efficiency, it is crucial to study the effects of ED thresholds ($\rho_\tau$) and sampling point counts ($\xi$) on model performance. Figure 5 shows the ablation results of PointVector under varying $\rho_\tau$ and $\xi$. We observe that performance does not improve proportionally with decreasing $\rho_\tau$ or increasing $\xi$, highlighting the importance of carefully selecting these hyperparameters to strike a balance between accuracy and computational cost. See Appendix A.20 for more details. In addition, we also performed ablation experiments on the Bohr grid spacing (Appendix A.21) and the InfoNCE temperature (Appendix A.22).

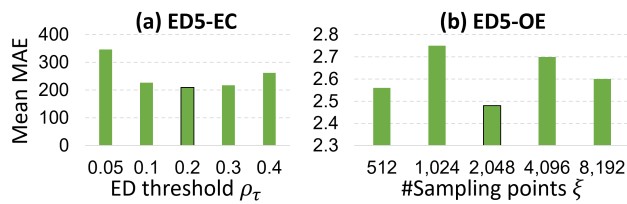

Figure 5: Ablation results of PointVector on (a) different ED thresholds $\rho_\tau$ and (b) different number of sampling points $\xi$.

## 5 Conclusion

In this work, we explore extending molecular and atomic-level modeling in machine learning force fields (MLFFs) to the electronic level by leveraging electron density (ED). We first construct EDBench, the largest-scale dataset of its kind to date, containing 3,359,472 molecular ED computed with density functional theory (DFT). To comprehensively evaluate the potential of ED-based molecular representations, we design three categories of medium-scale benchmark tasks on top of EDBench: prediction of diverse quantum chemical properties, retrieval between molecular structures (MS) and ED, and generation of ED from MS. Extensive experiments using several state-of-the-art models demonstrate the strong potential of ED as a representation for learning in the context of MLFFs.

**Limitations and future works.** Despite the significant progress made by the EDBench project in terms of the scale and quality of electron density (ED) data, there are still limitations. For instance, the choice of density functional can be further improved, and ED-specific models can be developed to represent ED from different perspectives, such as point clouds, voxels, or images. In retrieval tasks, advanced contrastive methods (e.g., MoCo or hard negatives) may further improve performance, especially when batch negatives are too easy. While the primary goal of this work is to highlight the role of ED in MLFFs and to provide a fair and reproducible benchmark, these limitations also point toward promising directions for future work. We plan to extend the dataset to include higher-level functionals and molecules relevant to materials science, and to develop advanced representation models tailored to ED, enabling EDBench to support a broader range of applications in physical and chemical sciences.

## 6 Acknowledgement

This work was supported by the National Natural Science Foundation of China (grant nos. U22A2037, 62425204, 62122025, 62450002, 62432011), Grants of Ningbo 2023CX050011.

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

# A Appendix

## A.1 Background Speed Reading: Electron Density, Machine-Learning Force Fields, and EDBench

### A.1.1 Definition of Electronic Density and Importance of ED Data at Scale

Electron density $\rho(r)$ is the quantum-mechanical probability of finding any electron in an infinitesimal volume around the point r in three-dimensional space. Because the Hohenberg-Kohn theorem proves that the ground-state energy, bond lengths, dipole moment, reactivity and essentially every other equilibrium property of a molecule are uniquely determined by $\rho(r)$, knowing the density is equivalent to knowing the molecule's chemistry. Yet obtaining $\rho(r)$ from first-principles Density Functional Theory (DFT) demands minutes to hours even for a small molecule; With 3.4 million molecules, the total computational cost exceeded 205,000 core-hours, equivalent to approximately 23.4 years of single-core compute time. Consequently, EDBench therefore provides the first large-scale resource in which every one of 3.36 million of molecule structures is accompanied by a consistent, high-level $\rho(r)$ grid, together with standard quantum observables such as total energy, dipole moments and frontier orbital energies, enabling machine-learning models to learn directly from the electron cloud.

### A.1.2 Description of Machine-Learning Force Fields

Chemists and materials scientists traditionally predict how molecules move, react or bind by solving Newton's equations with forces obtained from expensive quantum calculations. A machine-learning force field (MLFF) replaces this quantum step with a neural network that learns to predict forces or energies directly from atomic coordinates. Current MLFFs typically represent molecules as collections of atoms linked by bonds, angles and torsions, but they ignore the underlying electron distribution; this limits their accuracy whenever subtle electronic effects such as charge transfer or catalytic activation are involved.

### A.1.3 Application of EDBench in the Community

Researchers can pre-train encoders on the proposed large-scale density data and then fine-tune them on smaller downstream tasks such as property prediction, docking or reactivity estimation; Predictive models can learn the mapping from electron density to various quantum chemical properties, such as energy and orbital energies, enabling more efficient quantum chemical property predictions. Generative models can learn a direct map from structure to $\rho(r)$ to obtain DFT-quality densities in milliseconds; Cross-modal retrieval allows an unknown experimental density—obtained, for example, from X-ray or electron diffraction—to be matched automatically against the database for inverse design. In addition, $\rho(r)$ is physically meaningful, any model that operates on it may remain chemically interpretable: one can ask where the lone pairs reside or which regions are electron-rich and receive an physically grounded answer.

## A.2 Detailed introduction of DFT

### A.2.1 Development of quantum mechanics

The fundamental equation of quantum mechanics, the Schrödinger equation [25], serves as the theoretical foundation for describing many-electron systems. The Schrödinger equation can be expressed as

$$\hat{H}\psi = E\psi \tag{S1}$$

Among them, Ĥ is the Hamiltonian operator, $\psi$ is the many-electron wave function, and $E$ is the total energy of the system. Solving the multi-electron Schrödinger equation directly is challenging due to the complexity arising from the multivariable wave function and electron interactions. Consequently, early researchers introduced various wave function-based approximation methods to simplify the problem. For instance, the Born–Oppenheimer approximation assumes that the nuclei are stationary relative to the electrons, which simplifies the Schrödinger equation into a function of electronic variables [26]. The Hartree-Fock method further simplifies the many-electron problem into a single-electron problem by assuming that each electron moves independently in the average potential field formed by the other electrons [27]. Although the Hartree-Fock method significantly simplifies the calculations, it neglects electron correlation, leading to insufficient accuracy in some cases [70]. In

addition, the Hartree-Fock method scales with the number of electrons $n$ as $\mathcal{O}(n^4)$, its computational cost remains prohibitive for large polyatomic molecules. Density Functional Theory (DFT) is more suitable for large molecules and complex systems due to its lower computational cost ($\mathcal{O}(n^3)$) and its ability to incorporate electron correlation effects [28].

### A.2.2 Composition of effective electron potential energy

The basis of DFT is Hohenberg-Kohn theorem, and Kohn-Sham equation is the practical application form of DFT. In Kohn-Sham equation, $V_{\text{eff}}(r)$ is the effective single-electron potential energy, defined as

$$V_{\text{eff}}(r) = V_{\text{ext}}(r) + V_{\text{H}}(r) + V_{\text{xc}}(r) \tag{S2}$$

The external potential $V_{\text{ext}}(r)$ is typically provided by the atomic nuclei. $V_{\text{H}}(r)$ is the Hartree potential, which is represented by the convolution of the ED with the Coulomb kernel. The exchange-correlation potential $V_{\text{xc}}(r)$ is the variational derivative of the exchange-correlation energy functional.

### A.2.3 Selection of functional and basis set

To solve the equation S2, it is usually necessary to select the basis set, pseudopotential, and exchange correlation functional. The basic set includes plane wave method, numerical atomic orbital method, and augmented wave method. Norm-conserving pseudopotential (NCPP), ultrasoft pesudopotential (USPP) and projector augmented wave (PAW) are common pseudopotential methods. The exchange-correlation energy functional includes the Local Density Approximation (LDA) [30], the Generalized Gradient Approximation (GGA) [31], and hybrid functionals (such as B3LYP) [32]. In this paper, the exchange-correlation functional used is B3LYP, and the 6-31G**/+G** basis set is selected for combination. B3LYP integrates the advantages of the Hartree-Fock method and DFT. The 6-31G**/+G** basis set enhances computational accuracy by splitting the valence electron orbitals into two sets of basis functions and further incorporating diffuse functions. This combination achieves a great balance between precision and efficiency, making it more suitable.

### A.3 Quantum datasets overview: physical and chemical properties

In Table S1, we not only highlights the large-scale electron density annotations provided by EDBench but also visually and statistically contrasts its physical and chemical diversity with other datasets.This addition aims to offer domain experts a clearer perspective on how EDBench complements and extends the scope of current resources, emphasizing its distinct characteristics and broad coverage. We believe this enhanced visualization will be instrumental in understanding its potential applications in quantum chemistry research.

### A.4 Why not reuse the calculation results of PubCheMQC?

PCQM4Mv2 used in EDBench is a quantum chemistry dataset originally curated under the PubChemQC project[59], yet PubChemQC emphasizes ground-state electronic-structure quantities such as orbital energies[71], whereas our dataset centers on electronic density $\rho$. We provide CUBE-format $\rho$ files and NO occupancies that describe electron distribution in real space—data absent from PubChemQC. This enables direct analysis of bonding, charge transfer, and intermolecular interactions, making EDBench the large-scale, high-quality electronic-density resource PubChemQC lacks. In addition, compared to 6-31G* basis set on PubChemQC, EDBench employs the 6-31G/+G basis set with the higher computational accuracy, which adds polarization functions for hydrogen atoms and includes diffuse functions. Weak interactions, such as van der Waals forces, hydrogen bonds, and $\pi - \pi$ stacking, can be effectively described. This is more suitable for systems with high chemical accuracy requirements, such as anions and weakly interacting systems. Overall, EDBench provides the electron density resources that PubchemQC lacks and improves the DFT calculation settings.

### A.5 Example of ED visualization

Figure S1 illustrates the visualization of a molecule's electron density (ED) under varying threshold values $\rho_\tau$. A higher $\rho_\tau$ retains only regions with a higher probability of electron presence. When $\rho_\tau = 0$, all possible electron positions are preserved, resulting in a dense, cuboid-like distribution.

Table S1: Quantum datasets overview about physical and chemical properties.

| Datasets | EDBench | QM9 | QH9 | PubChemQC | QMugs | OQMD | ECD |
|---|---|---|---|---|---|---|---|
| **Moleculars** | 3359472 | 134K | 130K | 85M | 665k | 1.2M | 140K |
| **Conformers** | 3359472 | 134K | 130K | 85M | 2M | 1.2M | 140K |
| **Elements** | H, C, N, O, Ti, Ar, S, Se, He, Be, F, P, Si, Ca, Ga, Zn, Ge, Mg, B, Cl, As, Br | H, C, N, O, F | H, C, N, O, F | H, C, N, O, P, S, F, Cl, Na, K, Mg, Ca | H, C, N, O, P, S, Cl, K, Ca, Br, I | Inorganic crystals | Inorganic crystals |
| **Electron density $\rho$** | ✓(CUBE) | × | × | × | × | × | ✓(CHGCAR) |
| **Total Energy** | ✓ | × | × | ✓ | ✓ | ✓ | × |
| **7 Orbital energies (HOMO, LUMO)** | ✓ | ✓ | × | ✓ | ✓ | × | × |
| **HOMO-LUMO gap** | ✓ | ✓ | × | ✓ | ✓ | × | × |
| **Hamiltonians** | × | × | ✓ | × | ✓ | × | × |
| **Dipole Moment** | ✓ | ✓ | × | ✓ | ✓ | × | × |
| **Electronic Structure** | ✓ | × | × | ✓ | ✓ | ✓ | ✓ |
| **Other Energies** | DF-RKS Final Energy, Nuclear Repulsion Energy, One-Electron Energy, Two-Electron Energy, Exchange-Correlation Energy | Zero point vibrational energy, Internal energy at 0 K/298.15 K, Free energy at 298.15 K | - | - | Exchange-correlation energy, Nuclear repulsion energy, One-electron energy, Two-electron energy | Compound formation energies | - |

As $\rho_\tau$ increases, the number of ED points gradually decreases, and the molecular contour becomes more visually distinct.

It is worth noting that $\rho_\tau = 0$ leads to an overly dense ED representation, which poses challenges for both storage and computation. By tuning $\rho_\tau$, we can achieve a balance between ED information retention and computational efficiency. In our experiments, we adopt $\rho_\tau \leq 0.2$ as a practical choice. As shown in Figure S1, this setting significantly reduces the number of ED points while preserving the essential ED structural features of the molecule.

## A.6   Clarification of time costs

To clarify, the molecules in the EDBench dataset are based on DFT-optimized geometries, that is, they are optimized at the DFT level. Specifically, the molecular geometries were sourced from the PCQM4Mv2 dataset [59], where they were optimized using DFT at the B3LYP/6-31G* level. Detailed information about the DFT geometry optimization process can be found in reference [71]. Since the molecular geometries are already optimized at the quantum DFT level, we perform single-point calculations on these pre-optimized structures to obtain additional quantum chemical properties, such as electron density. This approach eliminates the need for full geometry optimization, and instead, each molecule undergoes just one SCF calculation, which is consistent with the time estimated for a single SCF cycle at the B3LYP/6-31G level on a single CPU. Thus, the 205,000 core-hours reported for 3.3 million molecules corresponds to the computational cost for these single-point SCF calculations.

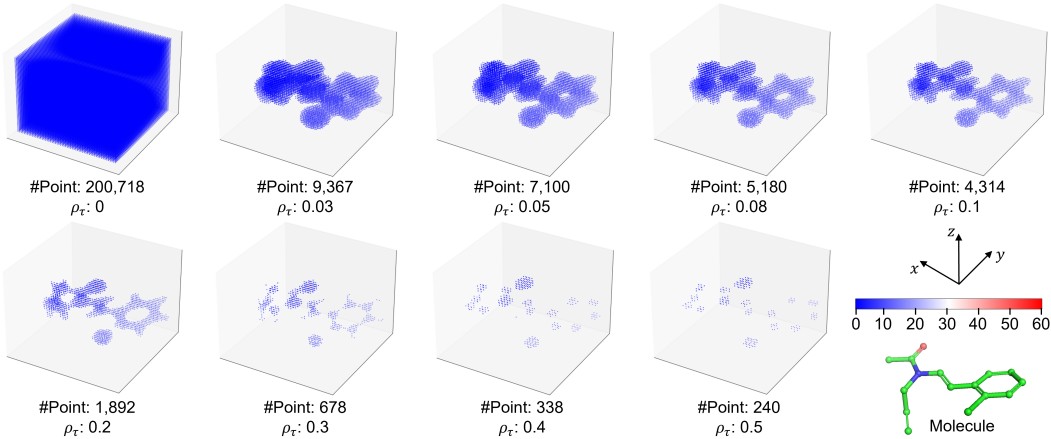

Figure S1: Example of ED visualization of a molecule with different thresholds $\rho_\tau$. #Point represents the number of ED points.

## A.7 Discussion on the Quality of the EDBench Database

To ensure the reliability and scientific utility of EDBench, we adopted a systematic and well-established protocol for electronic density (ED) calculation grounded in density functional theory (DFT) [28]. The entire workflow was designed to maximize both physical fidelity and computational robustness, while minimizing potential sources of error or bias.

First, all ED data were generated using Psi4 1.7, a widely used and validated open-source quantum chemistry package that supports high-accuracy ab initio and DFT calculations [60, 61, 56, 72]. We selected the B3LYP hybrid functional, a time-tested method known for its balance between computational efficiency and accuracy across a wide variety of molecules. Currently, B3LYP has been extensively applied in the domains of synthetic chemistry [73], molecular dynamics [74], phytochemistry [75], spectroscopy [76], medicine [56, 77] and physics [78]. This choice ensures that the resulting ED data reflect physically meaningful electron distributions rather than numerical artifacts.

Meanwhile, basis sets were systematically assigned based on molecular composition, employing 6-31G** for general cases and 6-31+G** for sulfur-containing molecules to capture diffuse electronic effects [79]. Compared with the basic 6-31G, 6-31G**/+G** provides the description of polarizable electron distribution and electron correlation effect by adding polarization function, which improves the processing ability of molecular polarization effect. This tailored approach enhances the accuracy of electron densities, particularly in chemically relevant regions such as lone pairs, $\pi$-systems, or polarizable atoms.

In addition, the reference wavefunction was selected according to spin multiplicity, with restricted Hartree-Fock (RHF) applied to closed-shell systems and unrestricted Hartree-Fock (UHF) used for open-shell systems [80], in line with Hund's rule. This guarantees correct treatment of $\alpha$ and $\beta$ spin components, reducing the risk of spin contamination and ensuring consistent modeling of open-shell species. To further control data quality, we enforced strict self-consistent field (SCF) convergence criteria before ED extraction [81, 82]. Electron density grids were then generated using a uniform grid spacing of 0.4 Bohr and a 4.0 Bohr padding, ensuring comprehensive spatial coverage without introducing undersampling or boundary artifacts. Additionally, a density fraction threshold of 0.85 was applied to focus on the physically relevant isosurface, filtering out low-density noise while preserving chemically meaningful features.

In sum, the quality of EDBench is supported by:

- A chemically sound and standardized computational protocol,
- Systematic and molecule-composition-based basis set selection,
- Accurate and consistent treatment of spin multiplicity,
- Rigorous convergence criteria and grid generation settings,

- Comprehensive spatial coverage and meaningful feature preservation.

In addition, post-calculation validation proceeded on two additional fronts. First, to verify that the choice of functional does not introduce systematic bias, we randomly selected 10,000 structures and recomputed their electron densities with the modern SCAN and $\omega$B97X-D functionals while keeping the basis set fixed. Experimental result demonstrates that B3LYP yields highly consistent electron density distributions compared to more modern functionals, with minimal loss in fidelity. Second, we examined the integrity of the neural-generated densities produced by HGEGNN (Section 4.4). Replacing the original DFT grids in the ED5-EC benchmark with these generated densities actually improved downstream model performance, indicating that the neural network not only preserved chemically relevant features but also introduced a beneficial regularisation effect. Reproducibility is guaranteed through complete open release. All data files, environment specifications, checkpoint and benchmarks from EDBench will be freely available.

These efforts collectively ensure that EDBench provides physically meaningful, reproducible, and high-resolution ED data at scale. We believe these safeguards sufficiently mitigate concerns of noise, bias, or low-quality samples, and position EDBench as a reliable benchmark for ED-aware machine learning research.

## A.8 Fidelity of different functionals to the electron density

We conducted a comparative evaluation of ED fidelity across different DFT functionals. Specifically, we randomly sampled 10,000 molecules from EDBench, and recomputed their electron densities using two higher-rung, modern functionals: SCAN and $\omega$B97X-D, while keeping the basis set consistent with B3LYP for a fair comparison. To ensure meaningful comparison, we first parsed the resulting cube files and interpolated all EDs onto a unified grid, aligning them to the grid resolution used in B3LYP calculations. We then computed voxel-wise Root Mean Square Error (RMSE) and Pearson correlation between the B3LYP densities and those obtained with SCAN and $\omega$B97X-D.

The results are summarized in Table S2. We observe that: (i) Between B3LYP and SCAN, the RMSE is 0.00039, with a Pearson correlation of 1.0; (ii) Between B3LYP and $\omega$B97X-D, the RMSE is 0.00022, also with a Pearson correlation of 1.0. These findings indicate that B3LYP yields highly consistent electron density distributions compared to more modern functionals, with minimal loss in fidelity. This validates our decision to use B3LYP in the construction of EDBench and further supports its scientific utility and reliability.

Table S2: Root Mean Square Error (RMSE) and Pearson Correlation coefficient between electron densities generated by different functionals.

| Functional Pair | RMSE ↓ | Pearson ↑ |
|---|---|---|
| B3LYP vs. SCAN | 0.00039±0.00011 | 1.000±0.000 |
| B3LYP vs. $\omega$B97X-D | 0.00022±0.00014 | 1.000±0.000 |

## A.9 Validation of electron densities for Ti- and Zn-containing molecules

Transition-metal-containing systems such as Ti and Zn require special consideration due to their being d-block elements and having partially filled d-orbitals. To validate the reliability of the basis set used in EDBench for these elements, we conducted a comprehensive evaluation comparing our current setup (B3LYP/6-31G**) against more advanced basis sets and functionals, including ECP-based options. Concretely, we extracted all Ti- and Zn-containing molecules from EDBench and recomputed their electron densities using 20 different functional-basis set combinations. These included 5 functionals (B3LYP, SCAN, $\omega$B97X-D, M06, and $\omega$B97M-V) and 4 basis sets (Def2-SVP, Def2-TZVP, Def2-TZVPP, and def2-QZVP), several of which are ECP-based and widely used for transition metal systems. We then compared the newly computed electron densities with those originally used in EDBench by computing voxel-wise RMSE and Pearson correlation. Full quantitative results are provided in Tables S3 and S4. In summary:

- The RMSE between EDBench densities and those from higher-level configurations ranged from 0.00091 (e.g., SCAN/Def2-TZVP and SCAN/Def2-TZVPP) to 0.00551 (e.g., B3LYP/Def2-SVP).

- The Pearson correlation coefficients across all comparisons were consistently near 1.0, indicating strong agreement in overall electron density structure.
- These findings demonstrate that the electron densities in EDBench—though generated using a moderate basis set—remain highly consistent with those produced by more advanced, ECP-based approaches, validating the practical quality and robustness of the dataset for Ti and Zn systems.

These findings demonstrate that the electron densities in EDBench—though generated using a moderate basis set—remain highly consistent with those produced by more advanced, ECP-based approaches, validating the practical quality and robustness of the dataset for Ti and Zn systems.

Table S3: Root mean square error (RMSE) between electron densities generated by different functional and basis set combinations (Functionals: B3LYP, SCAN, $\omega$B97X-D, M06, $\omega$B97M-V; Basis sets: def2-SVP, def2-TZVP, def2-TZVPP, def2-QZVP).

| RMSE ↓ | B3LYP | SCAN | wB97X-D | M06 | wB97M-V |
|---|---|---|---|---|---|
| Def2-SVP | 0.00551±0.00443 | 0.00162±0.00057 | 0.00545±0.00438 | 0.00349±0.00374 | 0.00543±0.00436 |
| Def2-TZVP | 0.00475±0.00397 | 0.00091±0.00015 | 0.00483±0.00406 | 0.00466±0.00384 | 0.00472±0.00395 |
| Def2-TZVPP | 0.00473±0.00394 | 0.00091±0.00015 | 0.00466±0.00389 | 0.00459±0.00378 | 0.00469±0.00392 |
| def2-QZVP | 0.00477±0.00394 | 0.00096±0.00015 | 0.00470±0.00388 | 0.00460±0.00376 | 0.00474±0.00393 |

Table S4: Pearson correlation coefficient between electron densities generated by different functional and basis set combinations (Functionals: B3LYP, SCAN, $\omega$B97X-D, M06, $\omega$B97M-V; Basis sets: def2-SVP, def2-TZVP, def2-TZVPP, def2-QZVP).

| Pearson ↑ | B3LYP | SCAN | wB97X-D | M06 | wB97M-V |
|---|---|---|---|---|---|
| Def2-SVP | 0.99993±0.00007 | 0.99999±0.00000 | 0.99993±0.00006 | 0.99997±0.00004 | 0.99993±0.00006 |
| Def2-TZVP | 0.99994±0.00005 | 1.00000±0.00000 | 0.99994±0.00005 | 0.99995±0.00005 | 0.99994±0.00005 |
| Def2-TZVPP | 0.99994±0.00005 | 1.00000±0.00000 | 0.99995±0.00005 | 0.99995±0.00005 | 0.99994±0.00005 |
| def2-QZVP | 0.99994±0.00005 | 1.00000±0.00000 | 0.99994±0.00005 | 0.99995±0.00005 | 0.99994±0.00005 |

## A.10 Statement on the maximum number of atoms

That most molecules in the current release of EDBench contain fewer than 20 heavy atoms. This design choice is deliberate and aligns with the PCQM4Mv2 distribution, which focuses on small, drug-like molecules. Prioritizing this molecular size range ensures both reliable DFT convergence and high chemical relevance for tasks such as property prediction and molecular generation.

While the current dataset emphasizes smaller molecules, EDBench already demonstrates broad elemental diversity by covering 22 distinct elements—significantly more than many existing datasets, as shown in Table 1. This highlights its potential for advancing modeling tasks beyond what traditional small-molecule benchmarks support.

Looking ahead, we plan to extend EDBench to include larger and more complex molecules, particularly those relevant to materials science and physical chemistry. To support this expansion, we will incorporate more advanced DFT functionals and element-specific basis sets to maintain accuracy and computational feasibility. We believe these enhancements will broaden the scope and applicability of EDBench across diverse scientific domains.

## A.11 Significance of benchmark tasks

We define three core tasks that capture distinct yet complementary capabilities of modeling electron density (ED), each grounded in both scientific motivation and real-world utility:

- **Prediction of quantum property.** As ED fundamentally determines molecular quantum behavior, predicting properties such as total energy, dipole moment, and orbital energies from ED allows us to assess whether a model has captured the underlying physical principles linking electron distributions to quantum observables. Despite ED being typically computed via expensive DFT simulations, it encodes richer quantum information than molecular geometry alone. Accurate property prediction from ED thus serves as a proxy for model

fidelity to quantum mechanics and offers a potential route to accelerate quantum property estimation in applications like drug discovery, catalysis, and materials design.

- **Retrieval between MS and ED.** Bidirectional retrieval between MS and ED enables molecule-level search in ED databases and supports structure inference from electronic environments. MS-to-ED retrieval facilitates functional site localization and electron distribution analysis, while ED-to-MS retrieval provides a foundation for inverse design driven by electronic requirements. This dual capability is essential for high-resolution virtual screening pipelines grounded in electronic behavior.

- **Generation of ED based on MS (Molecular Structure).** Learning to generate high-fidelity ED distributions directly from molecular structures bypasses the computational burden of DFT, making ED accessible to downstream tasks such as deep molecular dynamics, quantum-aware neural force fields, and reaction path modeling. This capability bridges the gap between computational efficiency and quantum-level accuracy, unlocking ED-driven learning for large-scale modeling scenarios.

### A.12 Detailed statistics of 6 benchmarks

We provide a detailed statistical analysis of six benchmarks in the EDBench suite: ED5-EC, ED5-OE, ED5-MM, ED5-OCS, ED5-MER, and ED5-EDP. Figures S2, S3, and S4 illustrate the distributions of the number of atoms, the number of ED points at the threshold $\rho_\tau = 0$, and the per-molecule mean ED values at $\rho_\tau = 0$, respectively. Specifically, ED length refers to the total number of ED sampling points retained after applying a density threshold $\rho_\tau$ (with $\rho_\tau = 0$ meaning all ED values are retained). This metric is analogous to the commonly used notion of molecular length (i.e., the number of heavy atoms), and serves to quantify the effective representational size of the ED modality. As shown, the number of ED points significantly exceeds the number of atoms, which provides richer information for force field learning and related downstream tasks.

Furthermore, we report the distribution of ED point counts and mean ED values under a higher threshold $\rho_\tau = 0.05$ in Figures S5 and S6, respectively. By applying a larger threshold (e.g., $\rho_\tau = 0.05$), the overall ED point count is significantly reduced, which can lead to improved computational efficiency. This suggests that threshold tuning offers a practical way to control the data volume without severely compromising structural fidelity. In addition, Figure S6 reveals that increasing the ED threshold implicitly forces the model to focus more on high-density regions, which are typically more chemically informative and relevant for modeling interactions.

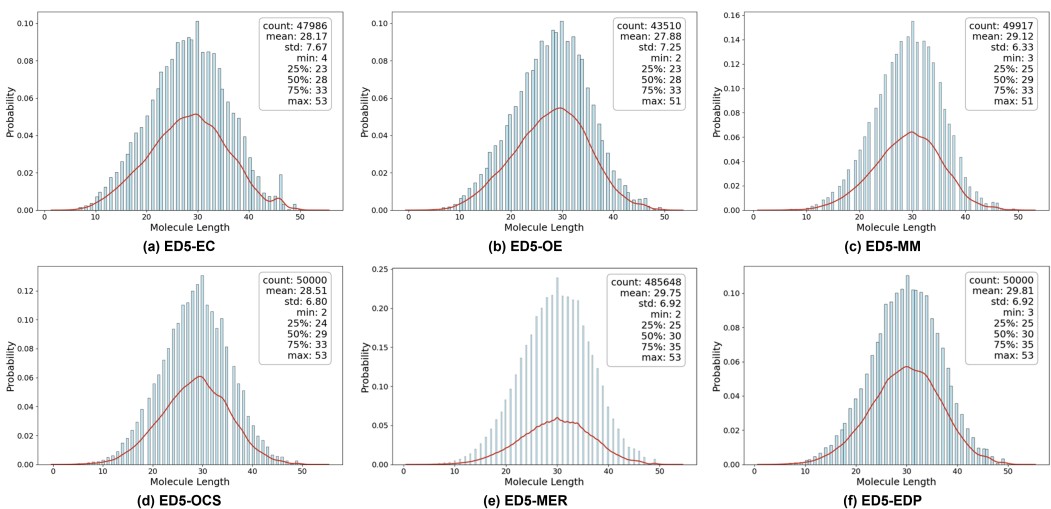

Figure S2: Distribution of the number of atoms in the 6 benchmark datasets.

### A.13 Details of computational efficiency

We conduct a computational efficiency analysis of all baseline models presented in this work, including molecular geometry-based methods—HGEGNN, EquiformerV2, and GeoFormer—and ED

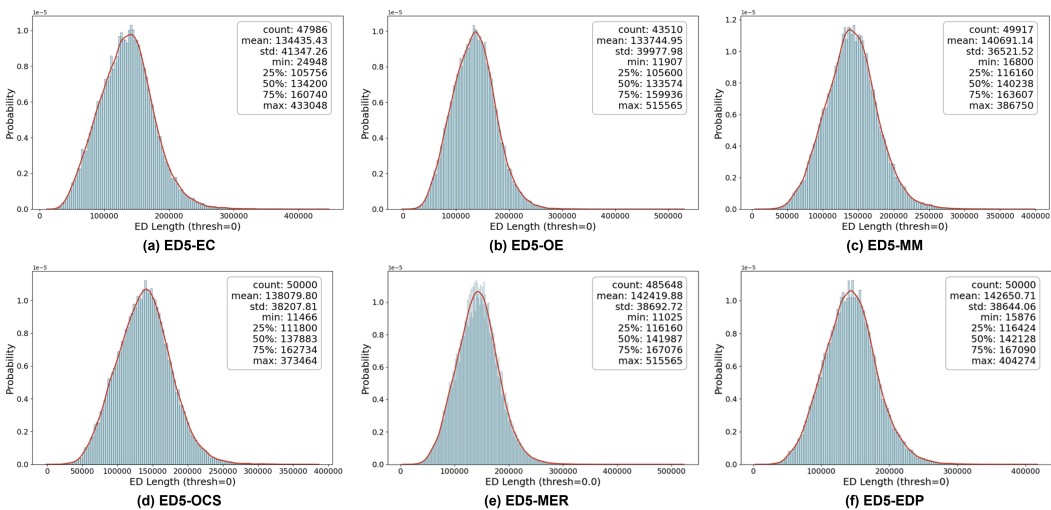

Figure S3: Distribution of the number of ED points in the 6 benchmark datasets with ED threshold $\rho_\tau = 0$.

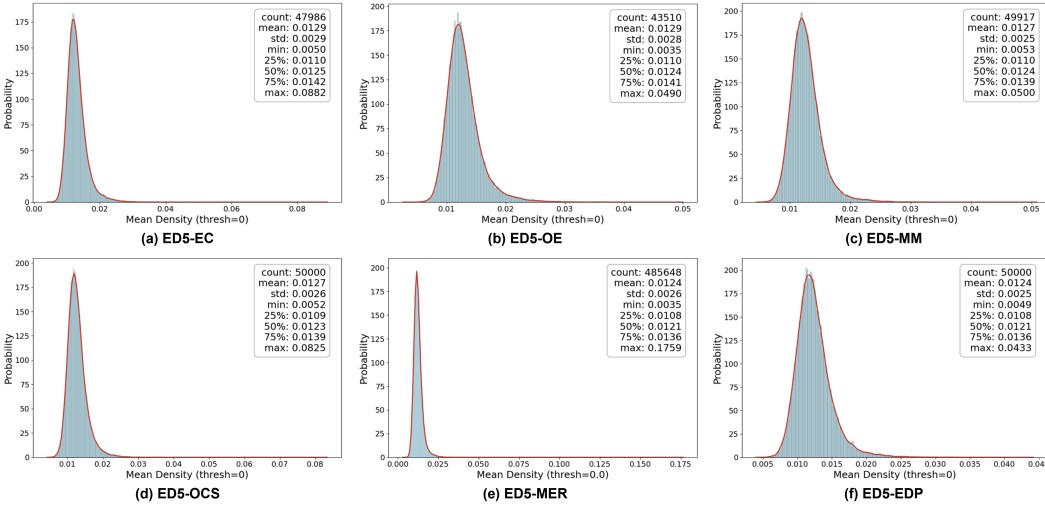

Figure S4: Distribution of per-molecule mean ED values in the 6 benchmark datasets with ED threshold $\rho_\tau = 0$.

point cloud-based methods—PointVector and X-3D. As a first step, we report the parameter count of each model to assess their relative model capacities. The details are summarized in Table S5. We find that the model sizes of EquiformerV2 and GeoFormer are significantly larger than the other models.

Table S5: The number of parameters of different models. #Params represents the number of parameters of the model. M stands for Million.

|  | HGEGNN | Equiformerv2 | GeoFormer | PointVector | X-3D |
|---|---|---|---|---|---|
| #Params (M) | 0.574 | 27.9 | 9.5 | 1.5454 | 0.9476 |

Next, we report the GPU memory usage and training time for each model. Due to varying memory requirements across models, we had to use different GPU devices to accommodate specific models and avoid out-of-memory (OOM) issues. Tables S6 and S7 present the computational efficiency of PointVector and X-3D, respectively. As expected, both GPU memory consumption and training time increase consistently with the number of sampling points $\xi$.

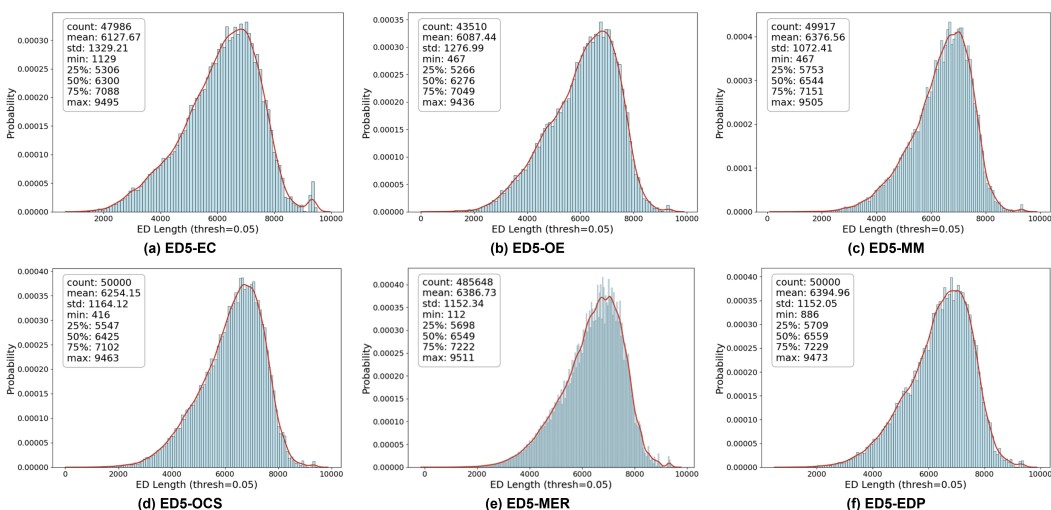

Figure S5: Distribution of the number of ED points in the 6 benchmark datasets with ED threshold $\rho_\tau = 0.05$.

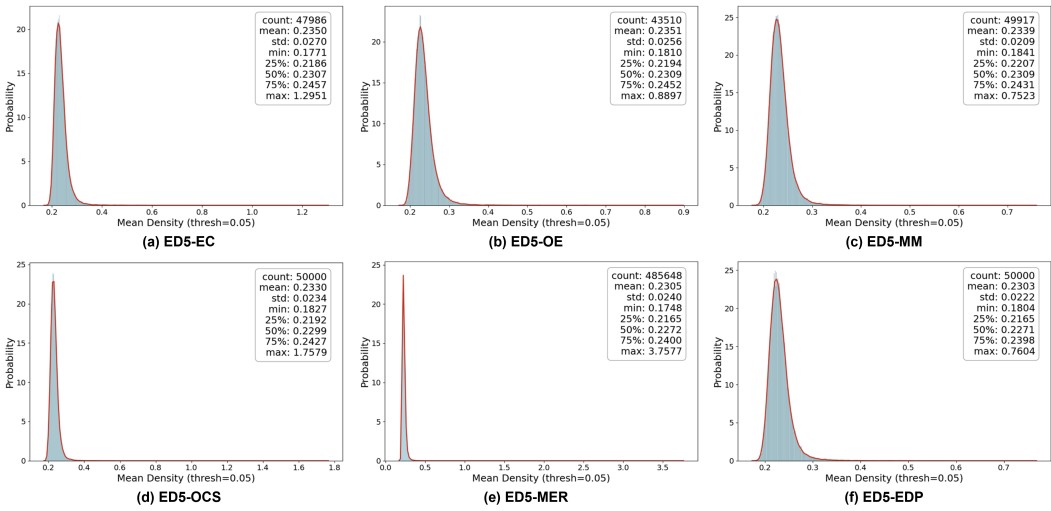

Figure S6: Distribution of per-molecule mean ED values in the 6 benchmark datasets with ED threshold $\rho_\tau = 0.05$.

Table S6: The computational efficiency of PointVector with different number of sampling points $\xi$ on ED5-OE dataset with batch size of 32 and epoch of 100. Time refers to the total time spent on the entire training process.

| $\xi$ | GPU Memory (MiB) | Time (minutes) | GPU |
|---|---|---|---|
| 512 | 4,425 | $\sim$100 | 3090 |
| 1024 | 6,623 | $\sim$150 | 3090 |
| 2048 | 11,453 | $\sim$325 | 3090 |
| 4096 | 20,757 | $\sim$433 | a100-80gb-pcie |
| 8192 | 38,083 | $\sim$850 | a100-80gb-pcie |

Table S7: The computational efficiency of X-3D with different number of sampling points $\xi$ on ED5-OE dataset with batch size of 32 and epoch of 100. Time refers to the total time spent on the entire training process.

| $\xi$ | GPU Memory (MiB) | Time (minutes) | GPU |
|---|---|---|---|
| 512 | 3,431 | $\sim$71 | 3090 |
| 1024 | 4,747 | $\sim$88 | 3090 |
| 2048 | 7,951 | $\sim$156 | 3090 |
| 4096 | 13,701 | $\sim$305 | 3090 |
| 8192 | 21,351 | $\sim$750 | 3090 |

Additionally, Table S8 shows the time efficiency of HGEGNN on the ED5-EDP dataset. A similar trend is observed: as the ED threshold $\rho_\tau$ decreases, the number of ED points increases, leading to higher memory usage and longer training times. These results collectively highlight the sensitivity of model efficiency to both the resolution of input data and the complexity of the architecture.

Table S8: The computational efficiency of HGEGNN with different ED threshold $\rho_\tau$ on ED5-EDP dataset. MiB/mol represents the total memory usage divided by the batch size.

| $\rho_\tau$ | GPU Memory (MiB/mol) | Time (minutes/epoch) | GPU |
|---|---|---|---|
| 0.1 | 2,153 | $\sim$15 | a100-80gb-pcie |
| 0.15 | 907 | $\sim$7.5 | a100-80gb-pcie |
| 0.2 | 616 | $\sim$5 | a100-80gb-pcie |

## A.14  Training with full corpus

Firstly, we clarify the usage of the EDBench corpus in our benchmark construction. In the ED5-MER task, 50,000 molecules serve as anchors. For each anchor, we sample 10 negative molecules, resulting in a total of 550,000 molecules involved in ED5-MER alone. After deduplication across all benchmark tasks, the entire benchmark suite involves approximately 680,000 unique molecules, accounting for roughly 20% of the full corpus.

Next, we have conducted additional large-scale training experiments. Specifically, we replace each task's training set with a pre-training corpus comprising about 2.67 million molecules not used in the benchmark. We use the pre-training corpus to train the model with an estimated training time of about two days, then evaluate it on the original validation and test sets of each task. Table S9 shows the performance of X-3D on ED5-OE. We observe that after training the model on the remaining large-scale dataset (X-3D (full)), its performance on the test set has been further improved. The results highlight the value of the full corpus for representation learning.

Table S9: MAE Performance of X-3D on ED5-OE After Training with Large-Scale Data and MAE$\times$100 metric.

| MAE | HOMO-2 | HOMO-1 | HOMO-0 | LUMO+0 | LUMO+1 | LUMO+2 | LUMO+3 |
|---|---|---|---|---|---|---|---|
| X-3D | 1.75$\pm$0.02 | 1.72$\pm$0.02 | 1.98$\pm$0.00 | 3.21$\pm$0.01 | 3.02$\pm$0.02 | 3.25$\pm$0.04 | 3.20$\pm$0.03 |
| X-3D (full) | 1.5797 | 1.6359 | 1.9104 | 2.9981 | 2.7028 | 2.8725 | 2.8708 |

## A.15  Effectiveness of extension to periodic systems

We conducted preliminary explorations to assess whether the data representations and model architectures proposed in EDBench can be effectively extended to periodic systems, such as crystalline solids.

**Dataset.** we extracted approximately 2,600 material molecules from the Materials Project (which primarily contains periodic systems, especially crystalline solids) and used the more suitable SCAN functional and Def2-SVP basis set to compute the electron density.

**Metrics.** We then split the dataset into training, validation, and test sets in an 8:1:1 ratio and evaluated it with tasks analogous to those in EDBench, including orbital energies prediction (EDMaterial-OE), multipole moment prediction (EDMaterial-MM), and electron density prediction from molecular structures (EDMaterial-EDP).

**Experimental results.** The experimental results are shown in Table S10 and Table S11. We find that X-3D and PointVector achieve strong performance on EDMaterial-OE and EDMaterial-MM, indicating that the proposed data representations and model architecture perform well in understanding material-based electron densities. This demonstrates that EDBench can be effectively extended to periodic systems, such as crystalline solids.

Table S10: MAE $\times 100$ performance of X-3D and PointVector on EDMaterial-OE task

| Model | HOMO-2 | HOMO-1 | HOMO-0 | LUMO+0 | LUMO+1 | LUMO+2 | LUMO+3 |
|---|---|---|---|---|---|---|---|
| X-3D | 3.45±0.02 | 3.47±0.01 | 3.21±0.02 | 3.41±0.02 | 4.04±0.02 | 4.39±0.04 | 4.53±0.07 |
| PointVector | 5.57±1.29 | 4.65±0.82 | 3.75±0.37 | 4.31±0.79 | 6.08±1.58 | 7.17±2.65 | 8.64±3.50 |

Table S11: MAE performance of X-3D and PointVector on EDMaterial-MM task.

| Model | Dipole X | Dipole Y | Dipole Z | Magnitude |
|---|---|---|---|---|
| **X-3D** | 0.8365±0.0098 | 1.0587±0.0280 | 1.2712±0.0577 | 1.3314±0.0187 |
| **PointVector** | 1.0568±0.0737 | 1.3476±0.0958 | 1.3615±0.0302 | 1.9749±0.0390 |

In addition, we find that DeepDFT is a structure-based method for generating electron densities and conduct further experiments using DeepDFT on both the ED5-EDP task and the newly constructed EDMaterial-EDP task. The results, presented in Table S12, show that DeepDFT achieved low MAE and high Pearson correlation for both ED5-EDP and EDMaterial-EDP. Notably, DeepDFT displayed lower performance in Spearman correlation, highlighting an area for future improvement.

Overall, these results further demonstrate the broader applicability of EDBench, especially in materials science, and we will continue to explore methods to improve performance, particularly in capturing non-linear relationships within electron-density fields.

Table S12: Performance of DeepDFT on EDMaterial-EDP task, showing MAE, Pearson, and Spearman correlations.

| Model | MAE | Pearson | Spearman |
|---|---|---|---|
| **ED5-EDP** | 0.018±0.003 | 0.993±0.004 | 0.381±0.162 |
| **EDMaterial-EDP** | 0.118±0.029 | 0.918±0.034 | 0.633±0.115 |

### A.16 Details of retrieval tasks (ED5-MER)

We first use GeoFormer and EquiformerV2 as molecular structure (MS) encoders, and PointVector and X-3D as electron density (ED) encoders. These encoders are combined pairwise—GeoFormer+PointVector, GeoFormer+X-3D, EquiFormer+PointVector, and EquiFormer+X-3D—to systematically evaluate cross-modal retrieval performance. Table S13 reports the Top-$k$ accuracy on both ED $\rightarrow$ MS and MS $\rightarrow$ ED tasks. Results reveal substantial performance differences among combinations. For example, GeoFormer+PointVector achieves only 17.67% Top-1 accuracy, while GeoFormer+X-3D reaches 68.32%, yielding an absolute improvement of 50.65%. Similarly, EquiFormer+PointVector achieves just 10.24% Top-1 accuracy, whereas EquiFormer+X-3D reaches 78.71%—an absolute gain of 68.47%. These results highlight the critical importance of selecting appropriate encoder architectures for effective cross-modal representation learning between MS and ED.

To further understand the performance gap, we closely analyzed the training logs of GeoFormer+PointVector and GeoFormer+X-3D. Figures S7(a) and S7(b) show their contrastive learning loss curves on the training and validation sets, respectively. While both combinations exhibit steadily decreasing training loss, GeoFormer+PointVector suffers from overfitting—as evidenced

Table S13: The Top-$k$ accuracy (%) on ED5-MER dataset. `ED → MS` represents using electron density (ED) to retrieve molecular structure (MS).

| MS model | ED model | ED → MS | | | MS → ED | | |
|---|---|---|---|---|---|---|---|
| | | Top-1 | Top-3 | Top-5 | Top-1 | Top-3 | Top-5 |
| GeoFormer | PointVector | 17.67±2.10 | 46.09±4.53 | 67.63±5.92 | 27.01±1.69 | 59.02±2.49 | 77.42±3.01 |
| | X-3D | 68.32±3.70 | 92.18±2.41 | 97.31±1.29 | 70.01±2.93 | 92.08±2.01 | 97.17±0.92 |
| EquiformerV2 | PointVector | 10.24±1.28 | 32.47±2.69 | 53.42±2.67 | 22.18±0.64 | 54.61±2.89 | 76.83±2.90 |
| | X-3D | **78.71±0.69** | **94.78±0.40** | **98.13±0.07** | **78.36±0.65** | **94.19±0.14** | **97.74±0.29** |

by its increasing validation loss despite continued improvement on the training set. In contrast, GeoFormer+X-3D maintains a consistently decreasing loss on both training and validation sets, explaining its significantly better retrieval performance.

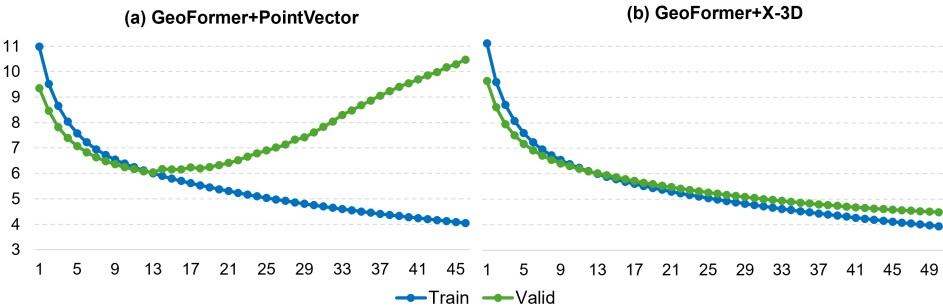

Figure S7: Comparative Learning Loss of GeoFormer+PointVector and GeoFormer+X-3D on ED5-MER training (Train) and validation (Valid) sets.

Overall, the strong bidirectional retrieval performance of GeoFormer+X-3D and EquiFormerV2+X-3D demonstrates the feasibility of learning the complex mapping between MS and ED, providing a solid foundation for retrieval-based applications. For example, retrieving the most compatible MS given an ED can enable a novel perspective on high-throughput virtual screening—particularly valuable in scenarios where the ED is known but the MS is unknown or ambiguous. Conversely, retrieving approximate ED distributions from MS opens a promising direction for building structure-driven, density-aware models, potentially enhancing the physical faithfulness of downstream tasks such as molecular property prediction and reactivity analysis.

### A.17 Sensitivity analysis on the hardness of negative samples in retrieval tasks

Table S14: Retrieval Performance of ED5-MER with EquiformerV2 and X-3D Across Different Negative Sample Hardness (Random, Easy, Hard).

| Split | ED→MS | | | MS→ED | | |
|---|---|---|---|---|---|---|
| | Top-1 | Top-2 | Top-3 | Top-1 | Top-2 | Top-3 |
| Random | **0.882 ± 0.005** | **0.959 ± 0.004** | 0.982 ± 0.001 | **0.882 ± 0.005** | **0.961 ± 0.001** | **0.983 ± 0.002** |
| Easy | 0.878 ± 0.004 | 0.957 ± 0.002 | **0.983 ± 0.002** | 0.873 ± 0.004 | 0.959 ± 0.001 | **0.983 ± 0.002** |
| Hard | 0.842 ± 0.009 | 0.947 ± 0.004 | 0.980 ± 0.001 | 0.836 ± 0.004 | 0.935 ± 0.005 | 0.971 ± 0.003 |

To explore how sensitive the ED5-MER performance is to the hardness of negative samples, we conducted retrieval experiments using a combination of the EquiformerV2 and X-3D models with negative samples of varying difficulty. Specifically, for each anchor, we used three types of negative samples, with five negative samples per type, meaning we needed to correctly retrieve the anchor from a set of six samples. The three types were as follows:

- **Random:** Negative samples were randomly selected from the remaining dataset.
- **Easy:** Negative samples were selected from a different cluster based on ECFP4 fingerprints and 3D USR descriptors (as detailed in the manuscript).
- **Hard:** Negative samples were selected from the same cluster as the anchor.

The experimental results are summarized in Table S14. We observed that the "random" setup yielded the highest retrieval performance, followed by "easy" and "hard" in that order. This suggests that the difficulty of the retrieval task increases from random to easy to hard, indicating that ED5-MER performance is sensitive to the hardness of negative samples.

## A.18  Error statistics for element-wise and size-resolved resolution

Here, we present element-wise and size-resolved error statistics, showing the MSE metrics for different electron density (ED) thresholds (0.1, 0.15, 0.2) across various atomic length ranges and atom types in the ED5-EDP task. As shown in Tables S15 and S16, we observe performance differences between these categories, which provide valuable insights into areas where the models may struggle. These results offer guidance for future research aimed at improving model performance in specific contexts.

Table S15: MSE metrics for different ED thresholds $\rho_\tau$ across atomic length ranges in the ED5-EDP task. MAX represents the maximum atomic length.

| #atoms | $\rho_\tau = 0.1$ | $\rho_\tau = 0.15$ | $\rho_\tau = 0.2$ |
|---|---|---|---|
| (0, 10) | 1.4533 | 0.1127 | 0.0617 |
| (10, 20) | 2.2329 | 0.1581 | 0.0657 |
| (20, 30) | 1.8257 | 0.0615 | 0.0394 |
| (30, 40) | 2.0581 | 0.271 | 0.1009 |
| (40, MAX) | 1.2474 | 0.0093 | 0.0251 |

Table S16: MSE metrics for different ED thresholds $\rho_\tau$ across atom types in the ED5-EDP task.

| atom type | $\rho_\tau = 0.1$ | $\rho_\tau = 0.15$ | $\rho_\tau = 0.2$ |
|---|---|---|---|
| H | 1.3387 | 0.0113 | 0.0262 |
| B | 0.6879 | 0.2677 | 2.069 |
| C | 1.0201 | 0.0063 | 0.0185 |
| N | 1.7475 | 0.0156 | 0.0234 |
| O | 3.0661 | 0.0551 | 0.0479 |
| F | 7.6878 | 0.1462 | 0.1472 |
| Si | 32.557 | 1.4758 | 0.7707 |
| P | 7.3819 | 2.8219 | 0.7615 |
| S | 27.2228 | 9.9254 | 3.5864 |
| Cl | 6.3195 | 0.7083 | 0.2215 |
| Ge | 85.3257 | 61.3484 | 3.5796 |
| Se | 3.3337 | 2.0152 | 0.5184 |
| Br | 85.2498 | 9.8684 | 8.6048 |

## A.19  Details about the quality analysis of ED outputs generated by generation task

To evaluate the quality of ED outputs in the generation task, we replace the DFT-based ED5-EC data (with a density threshold of $\rho_\tau = 0.2$) with new ED data generated by HGEGNN models trained on the original ED5-EC dataset. These new datasets are denoted as HGEGNN(2024), HGEGNN(2025), and HGEGNN(2026), where the numbers indicate different random seeds used during training. We then train PointVector—configured with the minimal ED length sampling rate—on each of these generated datasets. Detailed results are shown in Table S17. Compared to PointVector trained on the DFT-based ED5-EC, PointVector models trained on HGEGNN(2024)-, HGEGNN(2025)-, and HGEGNN(2026)-based ED5-EC all achieve superior performance. These findings support the feasibility of using deep learning models to accelerate DFT-level computations and suggest that the generated data is more learnable, thereby improving downstream model performance.

Table S17: MAE performance of PointVector on DFT-based and HGEGNN-generated ED5-EC datasets with $\rho_\tau = 0.2$. 2024, 2025, 2026 represent seeds of training HGEGNN on original ED5-EC dataset.

| | E1 | E2 | E3 | E4 | E5 | E6 | Mean |
|---|---|---|---|---|---|---|---|
| DFT | 224.13±43.47 | 155.85±28.75 | 451.59±58.53 | 190.47±25.62 | 9.57±1.56 | 224.13±43.47 | 209.29 |
| HGEGNN (2024) | 195.48±2.77 | 137.69±11.48 | 408.40±10.61 | 172.98±7.30 | **8.25±0.12** | 195.48±2.77 | 186.38 |
| HGEGNN (2025) | 208.18±16.43 | 142.33±9.94 | 428.24±7.31 | 180.90±5.02 | 8.85±0.36 | 208.18±16.43 | 196.11 |
| HGEGNN (2026) | **190.37±2.50** | **128.61±3.79** | **408.33±3.40** | **170.47±2.34** | 8.35±0.04 | **190.36±2.50** | **182.75** |

## A.20 Details of ablation study on threshold $\rho$ and the number of sampling $\xi$

**Ablation Study on the ED Threshold $\rho_\tau$.** The ED threshold $\rho_\tau$ plays a critical role in representing electron density, as it governs the trade-off between model performance and computational efficiency. In this ablation study, we evaluate the impact of $\rho_\tau$ using the PointVector model with a default number of sampled points $\xi = 2048$. However, when $\rho_\tau$ exceeds 0.05, the total number of ED points in some molecules falls below 2048, causing PointVector to fail due to insufficient input length. To minimize modifications to the original PointVector implementation, we set $\xi$ to the minimum ED length across the dataset. Table S18 shows the MAE of PointVector on the ED5-EC dataset under various ED thresholds. The results indicate that the best performance is achieved at $\rho_\tau = 0.2$, with an average MAE of 209.29. This demonstrates that tuning $\rho_\tau$ can effectively balance accuracy and computational cost.

Table S18: Ablation study (ED5-EC dataset) of PointVector on ED threshold $\rho_\tau$ with MAE metric.

| $\rho_\tau$ | $\xi$ | **E1** | **E2** | **E3** | **E4** | **E5** | **E6** | **Mean** |
|---|---|---|---|---|---|---|---|---|
| 0.05 | 2048 | 243.49±74.72 | 325.65±160.17 | 858.77±496.74 | 389.24±217.51 | 17.54±10.85 | 243.49±74.73 | 346.36 |
| 0.1 | 716 | 187.29±7.78 | 189.33±73.71 | 548.92±113.00 | 239.12±63.24 | 12.08±3.59 | 187.29±7.78 | 227.34 |
| 0.2 | 218 | 224.13±43.47 | 155.85±28.75 | 451.59±58.53 | 190.47±25.62 | 9.57±1.56 | 224.13±43.47 | **209.29** |
| 0.3 | 66 | 197.77±6.97 | 179.51±15.69 | 501.98±51.22 | 218.29±19.69 | 9.73±0.69 | 197.76±6.97 | 217.51 |
| 0.4 | 28 | 188.67±2.60 | 233.84±14.46 | 666.75±111.88 | 282.53±24.46 | 12.91±1.96 | 188.07±3.16 | 262.13 |

**Ablation Study on the Number of Point Cloud Samples $\xi$.** Point cloud-based methods (e.g., PointVector and X-3D) commonly adopt farthest point sampling (FPS) [83] to reduce the number of input points. Therefore, the number of sampled points, denoted as $\xi$, is a critical hyperparameter that directly affects both the model's capacity to capture spatial structures and its computational efficiency. A larger number of points allows the model to better represent the geometric details of ED, particularly in regions with ambiguous boundaries or sharp density gradients, facilitating the learning of fine-grained spatial features. However, increasing $\xi$ also leads to higher memory consumption and longer training and inference times, especially when dealing with large-scale ED datasets. Therefore, choosing an appropriate number of points is essential to balance representational power and computational cost. To investigate this trade-off, we evaluate the performance of the point cloud-based PointVector model under different sample sizes $\xi = \{512, 1024, 2048, 4096, 8192\}$. Table S19 reports the results on the ED5-OE dataset. We observe that PointVector achieves the best performance when $\xi = 2048$, reaching an average MAE of 0.0248. Additionally, the model performance does not monotonically improve with increasing $\xi$. This may be attributed to the model's limited capacity—PointVector contains only 1.5454M parameters—which may constrain its ability to effectively leverage a large number of ED points. This observation highlights the need for more strong and ED-specialized architectures in future work.

Table S19: Ablation study (ED5-OE dataset) of PointVector on the number of sampling points $\xi$ with MAE×100 metric and $\rho_\tau = 0.05$.

| $\xi$ | **HOMO-2** | **HOMO-1** | **HOMO-0** | **LUMO+0** | **LUMO+1** | **LUMO+2** | **LUMO+3** | **Mean** |
|---|---|---|---|---|---|---|---|---|
| 512 | 1.78±0.01 | 1.75±0.01 | 2.00±0.00 | 3.15±0.02 | 2.94±0.02 | 3.17±0.01 | 3.14±0.02 | 2.56 |
| 1024 | 1.79±0.01 | 1.74±1.98 | 3.19±2.99 | 3.19±0.02 | 2.99±0.02 | 3.19±0.02 | 3.14±0.01 | 2.75 |
| 2048 | 1.73±0.01 | 1.68±0.01 | 1.92±0.01 | 3.08±0.05 | 2.86±0.05 | 3.05±0.02 | 3.01±0.02 | **2.48** |
| 4096 | 1.87±0.09 | 1.76±0.07 | 2.01±0.02 | 3.40±0.07 | 3.21±0.12 | 3.38±0.19 | 3.29±0.13 | 2.70 |
| 8192 | 1.82±0.03 | 1.78±0.03 | 1.99±0.03 | 3.17±0.22 | 2.96±0.21 | 3.23±0.27 | 3.22±0.24 | 2.60 |

## A.21 Details of ablation study on Bohr grid spacing

The choice of a fixed 0.4 Bohr grid spacing and density thresholding could potentially affect the representation of diffuse electron density, particularly around anionic species or Rydberg states. Therefore, we conducted an ablation study to evaluate the sensitivity of our models to different grid spacing settings, thereby strengthening the analysis in Section 4.5. Specifically, we generated additional versions of the ED5-OE dataset using three alternative grid spacings: 0.2, 0.3, and 0.5 Bohr. We then re-evaluated two representative models—PointVector and X-3D—on each of these datasets. The results, summarized in Table S20, show that PointVector achieves MAEs ranging from 0.0248 to 0.0277 (a maximum difference of 0.0029), and X-3D achieves MAEs between 0.0245 and 0.0259 (a maximum difference of 0.0014). These consistently narrow ranges indicate that both models are highly robust to the choice of grid spacing, with only marginal variations in MAE across the different resolutions.

This insensitivity can be attributed to two main factors: (i) Electron density changes smoothly in space, so even when a slightly coarser grid is used, the key local patterns in the density are still well preserved. As a result, grid-based representations can capture the essential features without significant loss of information; (ii) The downstream architectures (PointVector and X-3D) are designed to learn from spatial context and may be less reliant on fine-grained grid resolution once essential geometric and density features are retained. Moreover, while extremely diffuse states (e.g., Rydberg orbitals) may suffer from truncation in coarse grids, the EDBench primarily consists of compact, drug-like molecules where electron density is well localized. This further mitigates the impact of grid coarseness in the current setting.

Table S20: Ablation study of PointVector and X-3D on ED5-OE with varying Bohr grid spacing and MAE×100 metric.

| Model | Bohr | HOMO-2 | HOMO-1 | HOMO-0 | LUMO+0 | LUMO+1 | LUMO+2 | LUMO+3 | Mean |
|---|---|---|---|---|---|---|---|---|---|
| PointVector | 0.2 | 1.94±0.11 | 1.77±0.04 | 2.02±0.04 | 3.51±0.21 | 3.39±0.26 | 3.47±0.19 | 3.31±0.16 | 2.77 |
| | 0.3 | 1.80±0.03 | 1.74±0.03 | 1.97±0.03 | 3.23±0.11 | 3.05±0.13 | 3.24±0.04 | 3.14±0.02 | 2.60 |
| | 0.4 | 1.73±0.01 | 1.68±0.01 | 1.92±0.01 | 3.08±0.05 | 2.86±0.05 | 3.05±0.02 | 3.01±0.02 | **2.48** |
| | 0.5 | 1.84±0.05 | 1.78±0.07 | 2.00±0.06 | 3.37±0.23 | 3.17±0.25 | 3.35±0.22 | 3.23±0.15 | 2.68 |
| X-3D | 0.2 | 1.76±0.02 | 1.71±0.02 | 1.93±0.01 | 3.09±0.02 | 2.88±0.02 | 3.05±0.01 | 3.00±0.01 | 2.49 |
| | 0.3 | 1.73±0.02 | 1.68±0.01 | 1.92±0.01 | 3.04±0.01 | 2.81±0.01 | 3.00±0.02 | 3.00±0.03 | **2.45** |
| | 0.4 | 1.75±0.02 | 1.72±0.02 | 1.98±0.00 | 3.21±0.01 | 3.02±0.02 | 3.25±0.04 | 3.20±0.03 | 2.59 |
| | 0.5 | 1.76±0.01 | 1.74±0.01 | 1.99±0.01 | 3.21±0.01 | 3.01±0.01 | 3.21±0.01 | 3.16±0.00 | 2.58 |

## A.22 Details of ablation study on temperature $\tau$

Table S21: Retrieval performance of ED5-MER with EquiformerV2 and X-3D across different temperature $\tau = \{0.05, 0.07, 0.1, 0.25, 0.3, 0.5\}$.

| $\tau$ | ED→MS | | | MS→ED | | |
|---|---|---|---|---|---|---|
| | Top-1 | Top-3 | Top-5 | Top-1 | Top-3 | Top-5 |
| 0.05 | **0.796±0.006** | **0.952±0.004** | **0.982±0.002** | **0.790±0.003** | **0.947±0.003** | **0.980±0.002** |
| 0.07 | 0.787±0.007 | 0.948±0.004 | 0.981±0.001 | 0.784±0.007 | 0.942±0.001 | 0.977±0.003 |
| 0.1 | 0.782±0.009 | 0.943±0.005 | 0.977±0.002 | 0.782±0.006 | 0.939±0.004 | 0.976±0.001 |
| 0.25 | 0.776±0.002 | 0.939±0.001 | 0.975±0.001 | 0.775±0.003 | 0.937±0.002 | 0.972±0.001 |
| 0.3 | 0.776±0.007 | 0.938±0.001 | 0.974±0.002 | 0.778±0.004 | 0.935±0.003 | 0.971±0.001 |
| 0.5 | 0.775±0.002 | 0.940±0.001 | 0.976±0.000 | 0.777±0.002 | 0.934±0.001 | 0.970±0.001 |

In retrieval tasks, we used the fixed temperature $\tau = 0.07$ in the InfoNCE loss, we chose this value based on empirical evidence from prior works. Specifically, $\tau = 0.07$ has been shown to yield strong performance in contrastive learning frameworks [84, 64], and remains widely adopted in subsequent models [85, 86]. To assess robustness across temperature configurations, we conducted an ablation study varying $\tau \in \{0.05, 0.1, 0.25, 0.3, 0.5\}$, guided by values explored in prior literature [87, 88, 89, 90]. The experimental results are shown in Table S21. Overall, the retrieval performance across different temperatures is relatively consistent, with a maximum performance difference of just 2.1%. Additionally, we observed a clear trend: retrieval performance improves as the temperature decreases. For instance, in the ED→MS task, as the temperature decreases from 0.5 to 0.05, the

Top-1 performance steadily increases from 0.775 to 0.796. Therefore, we recommend using a lower temperature to maintain higher performance.

