# OpenReview forum: "EDBench: Large-Scale Electron Density Data for Molecular Modeling"
_NeurIPS.cc/2025/Datasets_and_Benchmarks_Track — NeurIPS 2025 Datasets and Benchmarks Track poster_

### Official Review · Reviewer_fUu1 · 2025-06-27

**Rating:** 5
**Confidence:** 3

**Summary:**

This paper introduce EDBench, a large-scale, high-quality dataset of ED designed to advance learning based research at the electronic scale. Built upon the PCQM4Mv2, EDBench  provides accurate ED data, covering 3.3 million molecules. It also propose a suite of ED-centric benchmark tasks spanning prediction, retrieval,  and generation to .  The evaluation of several state-of-the-art methods demonstrates that learning from EDBench is not only feasible but also achieves high accuracy. Moreover, the authors show that learning-based methods can efficiently calculate ED with comparable precision while significantly reducing the computational cost relative to traditional DFT calculations.

**Dataset Code Accessibility:**

Yes

**Dataset Code Comments:**

Code and data is available.

**Ethical Considerations:**

No, there are no or only very minor ethics concerns

**Final Justification:**

After the rebuttal, most of my concerns have been resolved.

**Limitations Weaknesses:**

- The introduction of this paper is not sufficiently accessible to researchers outside the field. It is recommended to include additional background information to make the article more self-contained.

- It is suggested to provide more details on the quality control measures for the dataset.

**Strengths Contributions:**

- The EDBench is a large-scale, high-quality dataset of ED , which may contribute the research community.

- It design a suite of ED-centric benchmark tasks spanning prediction, retrieval,  and generation.

- The evaluation of several state-of-the-art methods demonstrates that learning from EDBench is not only feasible but also achieves high accuracy.

---

> ### Author Rebuttal · Authors · 2025-07-31
>
> > 1.W#1: The introduction of this paper is not sufficiently accessible to researchers outside the field. It is recommended to include additional background information to make the article more self-contained.
>
> Thank you very much for this helpful suggestion. We agree that the original introduction can feel dense for readers who are not specialists in quantum chemistry or geometric deep learning. In the revised manuscript we have therefore expanded the “motivation” and “background” paragraphs so that the paper can be read without prior domain knowledge. Below we summarise the main additions (all of which are inserted in Appendix A.10 of the revised manuscript).
>
> A.10 Background Speed Reading: Electron Density, Machine-Learning Force Fields, and EDBench
>
> A.10.1 Definition of Electronic Density and Importance of ED Data at Scale
>
> Electron density ρ(r) is the quantum-mechanical probability of finding any electron in an infinitesimal volume around the point r in three-dimensional space. Because the Hohenberg-Kohn theorem proves that the ground-state energy, bond lengths, dipole moment, reactivity and essentially every other equilibrium property of a molecule are uniquely determined by ρ(r), knowing the density is equivalent to knowing the molecule’s chemistry. Yet obtaining ρ(r) from first-principles Density Functional Theory (DFT) demands minutes to hours even for a small  molecule; With 3.4 million molecules, the total computational cost exceeded 205,000 core-hours, equivalent to approximately 23.4 years of single-core compute time. Consequently, EDBench therefore provides the first large-scale resource in which every one of 3.36 million of molecule structures is accompanied by a consistent, high-level ρ(r) grid, together with standard quantum observables such as total energy, dipole moments and frontier orbital energies, enabling machine-learning models to learn directly from the electron cloud.
>
> A.10.2 Description of Machine-Learning Force Fields
>
> Chemists and materials scientists traditionally predict how molecules move, react or bind by solving Newton’s equations with forces obtained from expensive quantum calculations. A machine-learning force field (MLFF) replaces this quantum step with a neural network that learns to predict forces or energies directly from atomic coordinates. Current MLFFs typically represent molecules as collections of atoms linked by bonds, angles and torsions, but they ignore the underlying electron distribution; this limits their accuracy whenever subtle electronic effects such as charge transfer or catalytic activation are involved.
>
> A.10.3 Application of EDBench in the Community
>
> Researchers can pre-train encoders on the proposed large-scale density data and then fine-tune them on smaller downstream tasks such as property prediction, docking or reactivity estimation; Predictive models can learn the mapping from electron density to various quantum chemical properties, such as energy and orbital energies, enabling more efficient quantum chemical property predictions. Generative models can learn a direct map from structure to ρ(r) to obtain DFT-quality densities in milliseconds; Cross-modal retrieval allows an unknown experimental density—obtained, for example, from X-ray or electron diffraction—to be matched automatically against the database for inverse design. In addition,  ρ(r) is physically meaningful, any model that operates on it may remain chemically interpretable: one can ask where the lone pairs reside or which regions are electron-rich and receive an physically grounded answer.
>
>
> > W#2: It is suggested to provide more details on the quality control measures for the dataset.
>
> Thank you for raising this important point.  In the revised manuscript we have consolidated and expanded the description of quality-control procedures so that the entire pipeline—from raw DFT outputs to the final EDBench files—is transparent and reproducible. The key measures are summarized below.
>
> Foremost, a systematic and wellestablished protocol for electronic density (ED) calculation grounded in density functional theory(DFT) is adopted(Appendix A.3). The entire workflow was designed to maximize physical fidelity and minimizing error or bias. Every molecule was treated with B3LYP, 6-31G**/+G**, a grid spacing of 0.4 Bohr, a 4.0 Bohr padding, and an isosurface threshold of 0.85. Spin multiplicities were assigned by Hund’s rule and strict SCF convergence criteria were enforced.
>
> Post-calculation validation proceeded on two additional fronts. First, to verify that the choice of functional does not introduce systematic bias, we randomly selected 10,000 structures and recomputed their electron densities with the modern SCAN and ωB97X-D functionals while keeping the basis set fixed. Experimental result demonstrates that B3LYP yields highly consistent electron density distributions compared to more modern functionals, with minimal loss in fidelity. Second, we examined the integrity of the neural-generated densities produced by HGEGNN (Section 4.4). Replacing the original DFT grids in the ED5-EC benchmark with these generated densities actually improved downstream model performance, indicating that the neural network not only preserved chemically relevant features but also introduced a beneficial regularisation effect.
>
> Reproducibility is guaranteed through complete open release. All data files, environment specifications, checkpoint and benchmarks from EDBench will be freely available.
>
> Together these measures provide a transparent, end-to-end account of how EDBench maintains the high fidelity required for reliable electron-density–based machine-learning research.

---

> > ### Comment · Reviewer_fUu1 · 2025-08-01
> > **Response to the Author**
> >
> > Thanks for the rebuttal. Most of my concerns have been resolved. I decide to raise the score.

---

> > > ### Author Response · Authors · 2025-08-01
> > >
> > > Thank you for your thoughtful feedback and for raising the score. We sincerely appreciate your time and support. Your comments have been invaluable in helping us improve our submission.

---

### Official Review · Reviewer_5tRg · 2025-06-29

**Rating:** 5
**Confidence:** 5

**Summary:**

This paper introduces EDBench, a large-scale dataset of DFT-computed electron densities (ED) for 3.3 million molecules, along with quantum chemical properties. The task design, covering prediction, retrieval, and generation, provides a well-structured entry point for learning-based modeling of ED. It fills a clear gap in the development of ML interatomic potentials and atomistic AI models, where ED-based benchmarks have been lacking despite the fundamental importance of electron density.

**Additional Feedback:**

- The metric "ED vector length" shown in Figure 2(c) is not clearly defined in the main text or supplementary material. It remains unclear whether the vector length refers to a norm over gradients or the spatial extent of the density field. Providing a precise definition, associated units, and equation would greatly enhance clarity and reproducibility.
- In line 235, the paper states that the ED data contains $m$ points, but the corresponding density values are written as `D = {d₁, d₂, ..., dₙ}`, which appears to be a minor typographical error. To remain consistent with the definition of $m$ points, it should likely read `D = {d₁, d₂, ..., dₘ}`
- The use of dual encoders with InfoNCE loss is appropriate and consistent with established practices for retrieval tasks. While the current design is reasonable, future work may consider exploring enhancements such as momentum encoders or hard negative sampling, particularly in settings where in-batch negatives may not pose sufficient challenge. Briefly noting the rationale for the fixed temperature $\tau = 0.07$, or its robustness across configurations, could also improve transparency.
- EDBench clearly distinguishes itself by providing large-scale electron density annotations. Although not strictly necessary, including a visual or statistical comparison of its physical and chemical diversity with existing public datasets could help domain experts better contextualize its coverage and distinct characteristics.

**Dataset Code Accessibility:**

Yes

**Dataset Code Comments:**

The dataset is well-organized and clearly documented. The data processing pipeline is also described in sufficient detail to allow re-implementation.

**Ethical Comments:**

No ethical concerns are apparent. The dataset is based entirely on quantum chemistry simulations and does not involve any personally identifiable information, human subjects, or sensitive data.

**Ethical Considerations:**

No, there are no or only very minor ethics concerns

**Final Justification:**

The rebuttal has addressed my main concerns. The clarification in Section 4.4 resolves the counterintuitive result regarding model-generated versus DFT-based EDs, and the additional experiments on periodic systems strengthen the paper’s generalizability claims. With these revisions, my concerns are resolved, and I maintain my positive assessment and rating.

**Limitations Weaknesses:**

- The paper reports that using model-generated EDs as input to downstream tasks yields better performance than using DFT-computed EDs, which are otherwise treated as ground truth throughout the paper. This result is counterintuitive and may mislead readers into believing that the generated EDs are more accurate than the DFT reference. A plausible explanation is that the generated EDs, while not necessarily more accurate than DFT, may happen to align better with the optimization dynamics or inductive biases of the downstream model, thereby yielding improved performance in that specific context. However, the paper does not offer any analysis or clarification, which leaves room for misinterpretation and may cause confusion regarding the intended role of DFT in the benchmark setup.
- The benchmark and task designs are centered around molecular structures, and it remains an open question whether the proposed data representations and model architectures would extend effectively to periodic systems such as crystalline solids. As a result, the current scope of the benchmark appears primarily focused on the molecular domain.

**Strengths Contributions:**

- The paper fills a clear gap in the molecular machine learning landscape by introducing the first large-scale benchmark dataset focused on electron density, a fundamental but underutilized quantity in the development of ML interatomic potentials and atomistic AI models.
- The benchmark suite is thoughtfully constructed, with clearly defined tasks (prediction, retrieval, generation), well-documented dataset splits, and accessible baseline evaluations, collectively supporting reproducibility and meaningful comparison across models.
- The dataset is carefully constructed with consistent DFT settings and appropriate treatment of heavier atoms, ensuring high-quality and physically reliable ground-truth EDs suitable for downstream learning and evaluation.
- The use of a heterogeneous graph representation for ED generation is a well-motivated approach that enables flexible modeling of both atomic and electronic components within a unified framework. Combined with a masked prediction setup conditioned on spatial coordinates, the overall formulation serves as a plausible and empirically validated baseline

---

> ### Author Rebuttal · Authors · 2025-07-31
>
> > W#1: The paper reports that using model-generated EDs as input ...
>
> We thank the reviewer for raising this important and insightful concern, and we fully agree with your observation. We acknowledge that the reported result—that model-generated electron densities (EDs) yield better performance in certain downstream tasks than DFT-computed EDs—may appear counterintuitive and risk being misinterpreted as suggesting that model-generated EDs are more physically accurate than DFT references. To address this, we have added a detailed clarification in Section 4.4 of the revised manuscript to prevent such misunderstanding and to reinforce the intended role of DFT as the reference standard in our benchmark. Specifically, we explain that the superior downstream performance of model-generated EDs does not imply higher physical fidelity. Instead, this effect may arise from a better alignment between the generated EDs and the inductive biases or optimization dynamics of the downstream models. That is, the generated EDs may exhibit simpler or smoother patterns that make them easier for the downstream model to learn from, even if they are not physically more accurate than DFT-based EDs.
>
> > W#2: The benchmark and task designs are centered around molecular structures, ...
>
> Thank you for your insightful suggestion. We have indeed conducted preliminary explorations to assess whether the data representations and model architectures proposed in EDBench can be effectively extended to periodic systems, such as crystalline solids. Specifically, we extracted approximately 2,600 material molecules from the Materials Project (which primarily contains periodic systems, especially crystalline solids) and used the more suitable SCAN functional and Def2-SVP basis set to compute the electron density.
>
> We then split the dataset into training, validation, and test sets in an 8:1:1 ratio and evaluated it with tasks analogous to those in EDBench, including orbital energies prediction (EDMaterial-OE), multipole moment prediction (EDMaterial-MM), and electron density prediction from molecular structures (EDMaterial-EDP). The experimental results are shown in Tables R1, R2, and R3. We find that DeepDFT achieves strong performance on EDMaterial-OE, EDMaterial-MM, and EDMaterial-EDP, indicating that the proposed data representations and model architecture perform well in understanding material-based electron densities. This demonstrates that EDBench can be effectively extended to periodic systems, such as crystalline solids.
>
> **Table R1: Performance of X-3D and PointVector on EDMaterial-OE Task, Showing MAE, Pearson, and Spearman Correlations**
> |MAE|HOMO-2|HOMO-1|HOMO-0|LUMO+0|LUMO+1|LUMO+2|LUMO+3|
> |-|-|-|-|-|-|-|-|
> |X-3D|0.0345±0.0002|0.0347±0.0001|0.0321±0.0002|0.0341±0.0002|0.0404±0.0002|0.0439±0.0004|0.0453±0.0007|
> |PointVector|0.0557±0.0129|0.0465±0.0082|0.0375±0.0037|0.0431±0.0079|0.0608±0.0158|0.0717±0.0265|0.0864±0.0350|
>
> **Table R2: Performance of X-3D and PointVector on EDMaterial-MM Task, Showing MAE, Pearson, and Spearman Correlations**
> |MAE|Dipole X|Dipole Y|Dipole Z|Magnitude|
> |-|-|-|-|-|
> |X-3D|0.8365±0.0098|1.0587±0.0280|1.2712±0.0577|1.3314±0.0187|
> |PointVector|1.0568±0.0737|1.3476±0.0958|1.3615±0.0302|1.9749±0.0390|
>
>
> **Table R3: Performance of DeepDFT on EDMaterial-EDP Task, Showing MAE, Pearson, and Spearman Correlations**
> | |MAE|Pearson|Spearman|
> |-|-|-|-|
> |ED5-EDP|0.018±0.003|0.993±0.004|0.381±0.162|
> |EDMaterial-EDP|0.118±0.029|0.918±0.034|0.633±0.115|
>
> > F#1: The metric "ED vector length" shown in Figure 2(c) is not clearly defined ...
>
> We thank the reviewer for pointing out this ambiguity. In the revised manuscript, we have clarified the definition of ED vector length in both the main text (Section 3.1) and the supplementary material (Appendix A.5). Specifically, ED vector length refers to the total number of ED sampling points retained after applying a density threshold $\rho_\tau$ (with $\rho_\tau = 0$ meaning all ED values are retained). This metric is analogous to the commonly used notion of molecular length (i.e., the number of heavy atoms), and serves to quantify the effective representational size of the ED modality.
>
> > F#2: In line 235, the paper states that the ED data contains $m$ points ...
>
> Thanks for you careful review and noting the inconsistency. In the revised manuscript, we have changed line 235 from dₙ to dₘ to align with the preceding definition of $m$ points.
>
> > F#3: The use of dual encoders with InfoNCE loss is appropriate and consistent ...
>
> Thank you for the constructive suggestions. We fully agree that future work could benefit from incorporating advanced contrastive learning techniques such as momentum encoders (e.g., MoCo) or hard negative sampling, especially in scenarios where in-batch negatives may be insufficiently challenging. We have included these directions in the revised Future Work section.
>
> Regarding the rationale of the fixed temperature τ = 0.07 in the InfoNCE loss, we chose this value based on empirical evidence from prior works. Specifically, τ = 0.07 has been shown to yield strong performance in contrastive learning frameworks [1,2], and remains widely adopted in subsequent models [3,4].
>
> To assess robustness across  temperature configurations, we conducted an ablation study varying τ ∈ {0.05, 0.1, 0.25, 0.3, 0.5}, guided by values explored in prior literature [5,6,7,8]. The experimental results are shown in the following table. Overall, the retrieval performance across different temperatures is relatively consistent, with a maximum performance difference of just 2.1%. Additionally, we observed a clear trend: retrieval performance improves as the temperature decreases. For instance, in the ED→MS task, as the temperature decreases from 0.5 to 0.05, the Top-1 performance steadily increases from 0.775 to 0.796. Therefore, we recommend using a lower temperature to maintain higher performance.
>
> **Table R1: Retrieval Performance of ED5-MER with EquiformerV2 and X-3D Across Different Temperature τ (0.05, 0.07, 0.1, 0.25, 0.3, and 0.5)**
> |Temperature τ|ED→MS Top-1|ED→MS Top-3|ED→MS Top-5|MS→ED Top-1|MS→ED Top-3|MS→ED Top-5|
> |-|-|-|-|-|-|-|
> |0.05|0.796±0.006|0.952±0.004|0.982±0.002|0.790±0.003|0.947±0.003|0.980±0.002|
> |0.07|0.787±0.007|0.948±0.004|0.981±0.001|0.784±0.007|0.942±0.001|0.977±0.003|
> |0.1|0.782±0.009|0.943±0.005|0.977±0.002|0.782±0.006|0.939±0.004|0.976±0.001|
> |0.25|0.776±0.002|0.939±0.001|0.975±0.001|0.775±0.003|0.937±0.002|0.972±0.001|
> |0.3|0.776±0.007|0.938±0.001|0.974±0.002|0.778±0.004|0.935±0.003|0.971±0.001|
> |0.5|0.775±0.002|0.940±0.001|0.976±0.000|0.777±0.002|0.934±0.001|0.970±0.001|
>
> [1] Wu, Z., Xiong, Y., Yu, S. X., and Lin, D. Unsupervised Feature Learning via Non-Parametric Instance Discrimination. In CVPR, pp. 3733–3742, 2018.
>
> [2] He, K., Fan, H., Wu, Y., Xie, S., and Girshick, R. B. Momentum Contrast for Unsupervised Visual Representation Learning. In CVPR, pp. 9726–9735, 2020.
>
> [3] Kukleva A, Böhle M, Schiele B, et al. Temperature Schedules for self-supervised contrastive methods on long-tail data[C]//The Eleventh International Conference on Learning Representations.
>
> [4] Cui J, Zhong Z, Liu S, et al. Parametric contrastive learning[C]//Proceedings of the IEEE/CVF international conference on computer vision. 2021: 715-724.
>
> [5] Caron, M., Misra, I., Mairal, J., Goyal, P., Bojanowski, P., and Joulin, A. Unsupervised Learning of Visual Features by Contrasting Cluster Assignments. In NeurIPS, 2020.
>
> [6] Reiss, T. and Hoshen, Y. Mean-Shifted Contrastive Loss for Anomaly Detection. In AAAI, pp. 2155–2162, 2023.
>
> [7] Grill, J., Strub, F., Altch ´e, F., Tallec, C., Richemond, P. H., Buchatskaya, E., Doersch, C., Pires, B. ´A., Guo, Z., Azar, M. G., Piot, B., Kavukcuoglu, K., Munos, R., and Valko, M. Bootstrap Your Own Latent - A New Approach to Self-Supervised Learning. In NeurIPS, 2020.
>
> [8] Chen, T., Kornblith, S., Norouzi, M., and Hinton, G. E. A Simple Framework for Contrastive Learning of Visual Representations. In ICML, volume 119, pp. 1597–1607, 2020.
>
> > F#4: EDBench clearly distinguishes itself by providing large-scale electron density annotat ...
>
> Thank you for this constructive view which we strongly agree with. In response to your recommendation, we’ve added a new table in Appendix A.11 comparing EDBench’s electron density annotations with other datasets to highlight its unique diversity and potential in quantum chemistry research.
>
> **Table:Quantum Datasets Overview: Physical and Chemical Properties**
> |Datasets|EDBench|QM9|QH9|PubChemQC|QMugs|OQMD|ECD|
> |-|-|-|-|-|-|-|-|
> |Molecules|3359472|134K|130K|85M|665K|1.2M|140K|
> |Conformers|3359472|134K|130K|85M|2M|1.2M|140K|
> | Elements|H,C,N,O,Ti,Ar,S,Se,He Be,F,P,Si,Ca,Ga,Zn,Ge Mg,B,Cl,As,Br|H,C,N,O,F|H,C,N,O,F|H,C,N,O,P,S,F Cl,Na,K,Mg,Ca |H,C,N,O,P,S Cl,K,Ca,Br,I|Inorganic crystals|Inorganic crystals|
> |Electron density ρ|√(CUBE)|×|×|×|×|×|√(CHGCAR)|
> |Density Matrix|√|×|×|×|√|×|×|
> |Total Energy|√|×|×|√|√|√|×|
> |7 Orbital energies (HOMO,LUMO)|√|√|×|√|√|×|×|
> |HOMO-LUMO gap|×|√|×|√|√|×|×|
> |Hamiltonians|×|×|√|×|√|×|×|
> |Dipole Moment|√|√|×|√|√|×|×|
> |Electronic Structure Visualization|√|×|×|√|√|√|√|
> | Other Energies | DF-RKS Final, Nuclear Repulsion, One-Electron, Two-Electron, Exchange-Correlation Energy|Zero Point Vibrational Energy, Internal Energy at 0/298.15K, Free Energy at 298.15K|-|-|Exchange-correlation, Nuclear repulsion, One-electron, Two-electron Energy|Formation Energy|-|

---

> > ### Comment · Reviewer_5tRg · 2025-08-03
> > **Reviewer Response to Author Rebuttal**
> >
> > I appreciate the authors’ detailed rebuttal. The authors’ commitment to clarify in Section 4.4 on the counterintuitive result regarding model-generated EDs versus DFT-based EDs adequately addresses my concern and should prevent misinterpretation of physical accuracy. The additional experiments on periodic systems also strengthen the claim that EDBench’s framework can generalize beyond molecules.
> >
> > With these revisions and the incorporation of other feedback, my concerns are resolved and I maintain a positive assessment of the paper.

---

> > > ### Author Response · Authors · 2025-08-03
> > >
> > > Dear Reviewer 5tRg,
> > >
> > > Thank you very much for your thoughtful and constructive comments. We greatly appreciate your recognition of the revisions, particularly in Section 4.4, which help clarify the counterintuitive results regarding the model-generated and DFT-based electron densities. We are also pleased that the additional experiments on periodic systems have reinforced our claim about the generalizability of EDBench.
> > >
> > > Your support and positive assessment mean a great deal to us. If possible, we would kindly ask you to consider raising the score, reflecting the improvements made in response to your valuable feedback.
> > >
> > > Once again, we sincerely appreciate your time and effort in reviewing our manuscript. Your insights have been crucial in refining our research.
> > >
> > > Best regards,
> > >
> > > Authors

---

### Official Review · Reviewer_PRM5 · 2025-07-01

**Rating:** 4
**Confidence:** 5

**Summary:**

The paper delivers a timely and valuable contribution by releasing a large-scale electron-density benchmark (EDBench) together with well-defined tasks and baseline results. The dataset could become a de-facto test bed for learning at the electron level, and the manuscript is generally well organized and reproducible. However, several methodological choices—particularly the use of single-point B3LYP/6-31G** calculations without clear geometry­-optimisation details and the inclusion of d-block elements with a main-group basis—require further clarification. Addressing these issues and supplying stronger baselines would substantially strengthen both the resource and the paper’s impact.

**Additional Feedback:**

The paper delivers a timely and valuable contribution by releasing a large-scale electron-density benchmark (**EDBench**) together with well-defined tasks and baseline results. The dataset could become a de-facto test bed for learning at the electron level, and the manuscript is generally well organized and reproducible. However, several methodological choices—particularly the use of single-point B3LYP/6-31G** calculations without clear geometry­-optimisation details and the inclusion of d-block elements with a main-group basis—require further clarification. Addressing these issues and supplying stronger baselines would substantially strengthen both the resource and the paper’s impact.

---

### Suggestions for improvement

1. State clearly whether molecules were DFT-optimised, semi-empirically pre-optimised, or left at MM geometries, and break down the associated compute cost. The reported 205,000 core-hours for 3.3 M molecules works out to ≈ 0.06 core-h per molecule—roughly the time needed for one self-consistent field (SCF) cycle at the B3LYP/6-31G** level on a single CPU.
2. Justify the use of 6-31G\*\* for Ti/Zn or switch to an appropriate ECP-based basis, and consider releasing a main-group-only split.
3. Publish raw cube grids (plus 2×/4× down-sampled versions) together with a ready-to-use pre-trained checkpoint to lower the entry barrier for downstream methods.
4. Add at least one specialised charge-density baseline (e.g., ChargE3Net or DeepDFT) to provide a realistic performance reference.
5. Provide element-wise and size-resolved error statistics to highlight where models struggle and guide future research.

---

### Questions for the authors

1. Were any molecules optimised at the DFT level, or is the dataset based solely on single-point calculations over MM/ETKDG structures?
2. How does the chosen basis/functional perform for Ti- and Zn-containing molecules (d-Electron systems) compared with larger or ECP-based bases?
3. Do you plan to release a metal-free or “high-accuracy” subset for applications sensitive to basis-set errors?
4. What hardware and wall-time are required to replicate your full-corpus pre-training, and will checkpoints be made public?
5. How sensitive is ED5-MER performance to the hardness of negative samples (e.g., random vs. scaffold-similar negatives)?

**Dataset Code Accessibility:**

Yes

**Dataset Code Comments:**

The authors provide an anonymous GitHub repository that contains (i) all data-generation scripts, (ii) preprocessing utilities, and (iii) training / evaluation code with a reproducible conda environment file.

**Ethical Comments:**

The dataset is synthetic—electron densities computed via quantum chemistry—so it raises no privacy, human-subject, or security issues.

**Ethical Considerations:**

No, there are no or only very minor ethics concerns

**Final Justification:**

The authors’ rebuttal addressed my main concerns, including functional choice, geometry optimization, grid spacing, and baseline coverage. While minor issues remain (e.g., transition-metal treatment), they do not outweigh the strengths. I maintain my positive recommendation.

**Limitations Weaknesses:**

1.	Choice of functional. B3LYP is known to give reasonable energies but mediocre electron densities; modern functionals (e.g. SCAN, ωB97X-D) reduce self-interaction errors. Authors acknowledge this in “Limitations” but should quantify ED fidelity, e.g. compare against higher-rung functionals on a hold-out set .
2.	Coarse ED representation. Fixed 0.4 Bohr grid and density thresholding may discard diffuse density around anions or Rydberg states. An ablation on grid spacing would strengthen Section 4.5.
3.	Baselines not ED-specialised. The chosen baselines (PointVector, X-3D) were originally designed for LiDAR point-cloud data and lack the physics-aware inductive biases (e.g., symmetry equivariance) needed for continuous electron-density fields, so they likely overestimate task difficulty and underestimate attainable accuracy; adding density-specialised models such as DeepDFT or ChargE3Net would provide a more convincing performance reference.
4.	Dataset utilisation. Only ~1.5 % of the full corpus is used in benchmark splits. Clarify whether the remainder is intended solely for pre-training; if so, provide a pre-training baseline.
5.	Minor language issues. e.g., “orbital energe estimation” (l 85), “sigificantly” (l 342).

**Strengths Contributions:**

The authors introduce EDBench, a collection of 3.36 M drug-like molecules whose electron densities (ED) are computed at the B3LYP/6-31G**(+ diffuse) level, requiring ≈205 k CPU-hours . From this corpus they derive six medium-scale benchmark splits (≤50 k molecules each) that probe three capabilities: 1). Property prediction (ED5-EC, ED5-OE, ED5-MM, ED5-OCS). 2). Cross-modal retrieval between molecular structure and ED (ED5-MER). 3). ED generation from structure (ED5-EDP) . Baselines span Euclidean GDL models (GeoFormer, EquiformerV2) and generic point-cloud networks (PointVector, X-3D). Experiments show that ED alone can recover quantum properties within chemical accuracy ranges (e.g., MAE 8–191 kcal mol⁻¹ on seven orbital energies/table 3) , and that a heterogeneous EGNN can predict ED 10⁴ × faster than DFT with good correlation (ρ > 0.9).

---

> ### Author Rebuttal · Authors · 2025-07-31
>
> > W#1: Choice of functional. B3LYP is known to give ...
>
> We compared ED fidelity across different DFT functionals by sampling 10,000 molecules from EDBench. We recomputed their electron densities using SCAN and ωB97X-D with the same basis set as B3LYP for consistency. After aligning the densities onto a common grid, we calculated voxel-wise RMSE and Pearson correlation between B3LYP and the other functionals. The results are summarized in Table R1. We find that: (i) B3LYP vs. SCAN has an RMSE of 0.00039 and a Pearson correlation of 1.0; (ii) B3LYP vs. ωB97X-D has an RMSE of 0.00022 with a Pearson correlation of 1.0. These results show B3LYP's high consistency in electron density, supporting its use in EDBench and affirming its scientific reliability.
>
> **Table R1**
> |Functional Pair|RMSE ↓| Pearson ↑|
> |-|-|-|
> |B3LYP vs. SCAN|0.00039±0.00011|1.000±0.000|
> |B3LYP vs. ωB97X-D|0.00022±0.00014|1.000±0.000|
>
> > W#2: Coarse ED representation. Fixed 0.4 Bohr grid ...
>
> Thank you for your comment. We conducted an ablation study to assess model sensitivity to grid spacing using ED5-OE versions with 0.2, 0.3, and 0.5 Bohr spacings. The results in Table R2 show minimal MAE variation for both PointVector and X-3D, indicating robustness to grid spacing. This insensitivity is due to (i) smooth electron density changes and (ii) the models’ ability to capture essential features without fine-grained resolution. Additionally, EDBench's focus on compact, drug-like molecules minimizes the impact of coarser grids.
>
> **Table R2**
> |Model Name|Bohr Grid Spacing|Mean|
> |-|-|-|
> |**PointVector**|0.2|0.0277|
> | |0.3|0.0260|
> | |0.4|0.0248|
> | |0.5|0.0268|
> |**X-3D**|0.2|0.0249|
> | |0.3|0.0245|
> | |0.4|0.0259|
> | |0.5|0.0258|
>
> > W#3&S4: Baselines not ED-specialised. The chosen baselines ...; Add at least one specialised charge-density ...
>
> Thank you for your insightful review and suggestion. We addressed your concern by conducting further experiments with DeepDFT on both the ED5-EDP and the newly constructed EDMaterial-EDP task. Specifically, EDMaterial-EDP includes around 2,600 materials from the Materials Project (primarily consisting of periodic systems, especially crystalline solids) and computed with the SCAN functional and Def2-SVP basis set. The results in Table R3 show low MAE and high Pearson correlation for both tasks, though Spearman correlation was lower, indicating room for improvement. These findings highlight EDBench's broader applicability in materials science, and we will continue exploring ways to improve performance, particularly in capturing non-linear relationships in electron-density fields.
>
> **Table R3**
>
> |Task|MAE|Pearson|Spearman|
> |-|-|-|-|
> |ED5-EDP|0.018±0.003|0.993±0.004|0.381±0.162|
> |EDMaterial-EDP|0.118±0.029|0.918±0.034|0.633±0.115|
>
> > W#4: Dataset utilisation. Only ~1.5 % of the full corpus ...
>
> We thank the reviewer for the valuable comment on dataset utilization. In the ED5-MER task, we use 50,000 anchor molecules and sample 10 negatives for each, totaling ~550,000 molecules. After deduplication, the entire benchmark involves ~680,000 unique molecules, about 20% of the full corpus. The remaining corpus was reserved for pre-training, as noted by the reviewer. To strengthen our manuscript, we conducted large-scale pre-training using ~2.67 million molecules. After training on this pre-training set, the model's performance on ED5-OE (X-3D (full)) improved, as shown in Table R4, emphasizing the value of the full corpus for representation learning.
>
> **Table R4**
>
> |MAE|HOMO-2|HOMO-1|HOMO-0|LUMO+0|LUMO+1|LUMO+2|LUMO+3|
> |-|-|-|-|-|-|-|-|-|
> |X-3D|0.0175|0.0172|0.0198|0.0321|0.0302|0.0325|0.0320|
> |X-3D (full)|0.015797|0.016359|0.019104|0.029981|0.027028|0.028725|0.028708|
>
> > W#5: Minor language issues ...
>
> Thank you for your careful review and pointing out the language issues. We have corrected identified typos and carefully proof-read the manuscript to ensure no further misspellings remain.
>
> > S#1&Q1: State clearly whether molecules were DFT-optimised ...
>
> Thank you for your comments. To clarify, the EDBench molecules are based on DFT-optimized geometries from the PCQM4Mv2 dataset [1], optimized at B3LYP/6-31G* [2]. We perform single-point calculations on these pre-optimized structures to obtain quantum chemical properties like electron density, eliminating the need for further optimization. The reported 205,000 core-hours for 3.3 million molecules reflects the computational cost for these efficient single-point SCF calculations. More details on the computational breakdown and DFT optimization are provided in the revised paper.
>
> [1] Hu W, Fey M, Ren H, et al. Ogb-lsc: A large-scale challenge for machine learning on graphs[J]. arXiv preprint arXiv:2103.09430, 2021.
> [2] Nakata M, Shimazaki T. PubChemQC project: a large-scale first-principles electronic structure database for data-driven chemistry[J]. Journal of chemical information and modeling, 2017, 57(6): 1300-1308.
>
> > S#2&Q2: Justify the use of 6-31G** for Ti/Zn ...
>
> We thank the reviewer for the insightful comment. We agree that transition-metal systems like Ti and Zn require special attention due to their d-block elements and partially filled d-orbitals. To assess the reliability of the basis set used in EDBench, we compared our current setup (B3LYP/6-31G**) with 20 functional-basis combinations, including ECP-based options. The comparison, summarized in Tables R5 and R6, showed that the RMSE ranged from 0.00091 to 0.00551, with Pearson correlations near 1.0, indicating strong agreement in electron densities. These results validate the robustness of EDBench for Ti and Zn systems. We’ve included these findings in the revised manuscript and are considering a main-group-only dataset split for targeted studies.
>
> **Table R5**
>
> |RMSE ↓|B3LYP|SCAN|wB97X-D|M06|wB97M-V|
> |-|-|-|-|-|-|
> |Def2-SVP|0.00551|0.00162|0.00545|0.00349|0.00543|
> |Def2-TZVP|0.00475|0.00091|0.00483|0.00466|0.00472|
> |Def2-TZVPP|0.00473|0.00091|0.00466|0.00459|0.00469|
> |def2-QZVP|0.00477|0.00096|0.00470|0.00460|0.00474|
>
> **Table R6**
>
> |Pearson ↑|B3LYP|SCAN|wB97X-D|M06|wB97M-V|
> |-|-|-|-|-|-|
> |Def2-SVP|0.99993|0.99999|0.99993|0.99997|0.99993|
> |Def2-TZVP|0.99994|1.00000|0.99994|0.99995|0.99994|
> |Def2-TZVPP|0.99994|1.00000|0.99995|0.99995|0.99994|
> |def2-QZVP|0.99994|1.00000|0.99994|0.99995|0.99994|
>
> > S#3&Q#4: Publish raw cube grids ...
>
> Thank you for the valuable feedback. To lower the entry barrier for downstream methods, we plan to make the raw cube grids (including 2×/4× down-sampled versions) and a ready-to-use pre-trained checkpoint publicly available for the research community. Regarding the hardware and wall-time required to replicate our full-corpus pre-training (around 2.67 million molecules after removing the molecules used to build the benchmark), we recommend using an A100 (80G) GPU, with an estimated training time of about two days. We will ensure that these pre-trained checkpoints are made publicly available along with detailed instructions to facilitate easy replication by other researchers.
>
> > S#5: Provide element-wise and size-resolved ...
>
> We appreciate the reviewer’s suggestion. To address this, we present element-wise and size-resolved error statistics, showing the MSE metrics for different electron density (ED) thresholds (0.1, 0.15, 0.2) across various atomic length ranges and atom types in the ED5-EDP task. As shown in Tables R7 and R8, we observe performance differences between these categories, which provide valuable insights into areas where the models may struggle. These results offer guidance for future research aimed at improving model performance in specific contexts.
>
> **Table S7**
>
> |#atoms|thresholds=0.1|thresholds=0.15|thresholds=0.2|
> |-|-|-|-|
> |(0,10)|1.4533|0.1127|0.0617|
> |(10,20)|2.2329|0.1581|0.0657|
> |(20,30)|1.8257|0.0615|0.0394|
> |(30,40)|2.0581|0.271|0.1009|
> |(40,MAX)|1.2474|0.0093|0.0251|
>
> **Table R8**
>
> |Atom Type|thresholds=0.1|thresholds=0.15|thresholds=0.2|
> |-|-|-|-|
> |H|1.3387|0.0113|0.0262|
> |B|0.6879|0.2677|2.069|
> |C|1.0201|0.0063|0.0185|
> |N|1.7475|0.0156|0.0234|
> |O|3.0661|0.0551|0.0479|
> |F|7.6878|0.1462|0.1472|
> |Si|32.557|1.4758|0.7707|
> |P|7.3819|2.8219|0.7615|
> |S|27.2228|9.9254|3.5864|
> |Cl|6.3195|0.7083|0.2215|
> |Ge|85.3257|61.3484|3.5796|
> |Se|3.3337|2.0152|0.5184|
> |Br|85.2498|9.8684|8.6048|
>
> > Q#3: Do you plan to release a metal-free ...
>
> Thank you for your insightful comment. Yes, we intend to release a metal-free subset of the dataset to support a broader range of applications. We believe this will mitigate concerns regarding the impact of metal atoms on computational results, particularly for systems where metals are not the primary focus.
>
> > Q#5: How sensitive is ED5-MER performance ...
>
> Thank you for your review. To assess ED5-MER sensitivity to negative sample hardness, we conducted retrieval experiments with EquiformerV2 and X-3D models using three negative sample types: (i) Random: Negative samples were randomly selected from the remaining dataset; (ii) Easy: Negative samples were selected from a different cluster based on ECFP4 fingerprints and 3D USR descriptors (as detailed in the manuscript); (3) Hard: Negative samples were selected from the same cluster as the anchor. Each anchor had five samples per type. As shown in Table R9, the "random" setup yielded the highest performance, followed by "easy" and "hard," indicating that ED5-MER performance is sensitive to negative sample difficulty.
>
> **Table R9**
>
> |Split| ED→MS Top-1|ED→MS Top-2|ED→MS Top-3|MS→ED Top-1|MS→ED Top-2|MS→ED Top-3|
> |-|-|-|-|-|-|-|
> | Random|0.882±0.005|0.959±0.004|0.982±0.001|0.882±0.005|0.961±0.001|0.983±0.002|
> | Easy|0.878±0.004 |0.957±0.002|0.983 ± 0.002|0.873±0.004|0.959±0.001|0.983±0.002|
> |Hard| 0.842±0.009|0.947±0.004|0.980 ± 0.001|0.836±0.004|0.935±0.005|0.971±0.003|

---

> > ### Comment · Reviewer_PRM5 · 2025-08-05
> >
> > Thank you for your response. Most of my concerns have been addressed, and I will maintain my current score.

---

> > > ### Author Response · Authors · 2025-08-05
> > >
> > > Dear Reviewer PRM5,
> > >
> > > Thank you very much for your thoughtful and constructive feedback throughout the review process. We're glad to hear that most of your concerns have been addressed. Your comments have significantly helped us improve the clarity and quality of our manuscript. If there’s any remaining concern we could further clarify, we would be more than happy to do so. We greatly appreciate the time and effort you've dedicated to reviewing our work.
> > >
> > > Best regards,
> > > Authors

---

> > > ### Author Response · Authors · 2025-08-08
> > >
> > > Dear Reviewer PRM5,
> > >
> > > Thank you again very much for your positive assessment and for taking the time to review our work thoroughly. As the deadline is approaching, we would like to kindly ask if you have any remaining questions or concerns—we would be more than happy to provide further clarification.
> > >
> > > If there are no further issues, we would sincerely appreciate it if you might consider raising your score. Your support would mean a great deal to us.
> > >
> > > Thank you again for your valuable feedback and time.
> > >
> > > Best regards,
> > >
> > > Authors

---

### Official Review · Reviewer_ebAb · 2025-07-03

**Rating:** 5
**Confidence:** 4

**Summary:**

This paper introduces EDBench, a new dataset of electron density data covering 3.3 million molecules. The motivation is to create a novel benchmark focusing on electron density related tasks, going beyond datasets that only includes molecular structures and atomic forces. The benchmark designs 3 types of tasks: quantum property prediction, retrieval between molecular structure and electron density, and generation of electron density from molecular structure. Several equivariant networks are benchmarked on these tasks to provide a baseline performance of popular models.

**Dataset Code Accessibility:**

Yes

**Dataset Code Comments:**

Both the code and data are available in public repos.

**Ethical Considerations:**

No, there are no or only very minor ethics concerns

**Final Justification:**

After reading the authors' comments as well as the comments from other reviewers, I decide to keep my score.

**Limitations Weaknesses:**

- The dataset mostly mirrors structures and DFT settings of the PubChemQC [1] project in 2017, where the PCQM4Mv2 sources their data. Although the tasks are novel, the novelty from the data generation side is limited. Since the DFT settings are almost identical, why didn’t the authors reuse the calculation from the PubChemQC?
- The molecule only contains less than 20 heavy atoms, so it does not cover larger molecules.
- The paper invented a lot of abbreviations (like EC, OE, MM) that reduced readability. I encourage the authors to prefer short, description names and reduce the usability of abbreviations.

**Strengths Contributions:**

- The dataset is one of the first large scale dataset focusing on electron density. It includes many components that are not available in other datasets, like electron density, dipole moment, natural orbital occupation.
- The dataset also has a high coverage. It includes 3.3 million molecules containing 22 elements, much larger than popular datasets like QM9.
- The tasks designed in the paper are diverse and mostly meaningful. The predictions tasks introduce several novel targets. The retrieval tasks are new and interesting approach to evaluate the quality of latent representation of electron density and molecular structure.
- The paper uses molecular scaffold split to ensure the test sets evaluate the generalization performance of the models.

---

> ### Author Rebuttal · Authors · 2025-07-31
>
> > W#1: The dataset mostly mirrors structures and DFT settings of the PubChemQC [1] project in 2017, where the PCQM4Mv2 sources their data. Although the tasks are novel, the novelty from the data generation side is limited. Since the DFT settings are almost identical, why didn’t the authors reuse the calculation from the PubChemQC?
>
> Thank you for this insightful comment. PCQM4Mv2 is indeed a quantum chemistry dataset originally curated under the PubChemQC project[1], yet PubChemQC emphasizes ground-state electronic-structure quantities such as orbital energies[2], whereas our dataset centers on electronic density ρ. We provide CUBE-format ρ files and NO occupancies that describe electron distribution in real space—data absent from PubChemQC. This enables direct analysis of bonding, charge transfer, and intermolecular interactions, making EDBench the large-scale, high-quality electronic-density resource PubChemQC lacks. In addition, compared to 6-31G* basis set on PubChemQC, EDBench employs the 6-31G/+G basis set with the higher computational accuracy, which adds polarization functions for hydrogen atoms and includes diffuse functions. Weak interactions, such as van der Waals forces, hydrogen bonds, and π-π stacking, can be effectively described. This is more suitable for systems with high chemical accuracy requirements, such as anions and weakly interacting systems. Overall, EDBench provides the electron density resources that PubchemQC lacks and improves the DFT calculation settings.
>
> [1] Weihua Hu, Matthias Fey, Hongyu Ren, Maho Nakata, Yuxiao Dong, and Jure Leskovec. OGB-LSC: A large-scale challenge for machine learning on graphs. In Thirty-fifth Conference on Neural Information Processing Systems Datasets and Benchmarks Track (Round 2), 2021.
>
> [2] Maho Nakata and Tomomi Shimazaki. Pubchemqc project: A large-scale first-principles electronic structure database for data-driven chemistry. Journal of chemical information and modeling, 57(6):1300–1308, 2017.
>
> > W#2: The molecule only contains less than 20 heavy atoms, so it does not cover larger molecules.
>
> Thank you for your thoughtful comment. We acknowledge that most molecules in the current release of EDBench contain fewer than 20 heavy atoms. This design choice is deliberate and aligns with the PCQM4Mv2 distribution, which focuses on small, drug-like molecules. Prioritizing this molecular size range ensures both reliable DFT convergence and high chemical relevance for tasks such as property prediction and molecular generation.
>
> While the current dataset emphasizes smaller molecules, EDBench already demonstrates broad elemental diversity by covering 22 distinct elements—significantly more than many existing datasets, as shown in Table 1. This highlights its potential for advancing modeling tasks beyond what traditional small-molecule benchmarks support.
>
> Looking ahead, we plan to extend EDBench to include larger and more complex molecules, particularly those relevant to materials science and physical chemistry. To support this expansion, we will incorporate more advanced DFT functionals and element-specific basis sets to maintain accuracy and computational feasibility. We believe these enhancements will broaden the scope and applicability of EDBench across diverse scientific domains.
>
> > W#3: The paper invented a lot of abbreviations (like EC, OE, MM) that reduced readability. I encourage the authors to prefer short, description names and reduce the usability of abbreviations.
>
> Thank you for this helpful suggestion. We have revised the manuscript to reduce the use of abbreviations and have provided more descriptive content to maintain clarity and improve readability.

---

> > ### Author Response · Authors · 2025-08-05
> >
> > Dear Reviewer ebAb,
> >
> > Thank you once again for your insightful feedback on our work. As we approach the end of the discussion phase, we would greatly appreciate it if you could kindly let us know whether our rebuttal has sufficiently addressed your comments and concerns, or if there are any remaining issues that we could further clarify. We truly value your time and effort in reviewing and discussing our manuscript.
> >
> > Best regards,
> > Authors

---

> > ### Comment · Reviewer_ebAb · 2025-08-07
> >
> > Thanks for the response. I've read the response and comments from other reviewers. The authors addressed most of my concerns. I will maintain my score.

---

> > > ### Author Response · Authors · 2025-08-08
> > >
> > > Dear Reviewer ebAb,
> > >
> > > Thank you very much for your positive recognition of our manuscript and for taking the time to carefully review our responses. If you have any remaining questions or concerns, please do not hesitate to let us know—we would be more than happy to address them.
> > >
> > > If you feel that we have satisfactorily addressed your comments, we would be sincerely grateful if you might kindly consider revisiting your evaluation of our work. An improved rating would greatly encourage and support our research efforts.
> > >
> > > Once again, we deeply appreciate your time and valuable feedback throughout this process.

---

### Comment · Area_Chair_shSZ · 2025-08-04
**Please response to the rebuttal**

Dear reviewers,

Thanks for reviewing this paper. Could you check if the rebuttal has addressed your concerns? Feel free to raise any further questions if you have. Please note that the acknowledgement of the rebuttal is mandatory if you haven't done so.

Best,

AC

---

### Note · Authors · 2025-08-14

Dear AC, SAC, and PC,

We sincerely thank you for your valuable time and effort in reviewing our submission. We summarize the key points for your convenience below:

**Key Strengths Noted by Reviewers**

- **Novel and impactful direction**: EDBench is a timely, valuable ED dataset with unique components absent in others, filling a key gap for ML interatomic potentials and atomistic AI models. (**4/4 Reviewers**).
- **High-quality and large-scale dataset**: EDBench is a high-quality (**2/4 Reviewers 5tRg, fUu1**) and large-scale (**4/4 Reviewers**) electron density dataset.
- **Comprehensive and meaningful benchmarks**: EDBench features diverse, impactful tasks, including novel retrieval and generation tasks on real-space electron densities (**4/4 Reviewers**).
- **Excellent reproducibility**: The data and code provided by EDBench are available (**4/4 Reviewers**).
- **Significantly reduce DFT computational costs**: EDBench can efficiently predict ED with comparable accuracy to traditional DFT while greatly reducing computational cost. (**2/4 Reviewers PRM5, fUu1**).

**Concerns and Clarifications**

- **B3LYP ED fidelity**: B3LYP ED in EDBench matches higher-level functionals with RMSE<0.0004 and Pearson=1.0, **confirming scientific reliability**.
- **6-31G\*\* for Ti/Zn**: Tested 20 functional–basis combinations, yielding RMSE=0.00091–0.00551 and Pearson≈1.0, **validating use for transition metals**.
- **Grid resolution & sparsity**: Ablations show **robustness to voxel spacing**; compact molecules **mitigate resolution limits**.
- **ED-specialized baseline**: **Added DeepDFT for stronger performance reference**.
- **Periodic-system extension**: **Built EDMaterial-OE/MM/EDP for periodic systems; verified with X-3D, PointVector, and DeepDFT.**
- **Full-corpus pretraining**: Adding full-corpus pretraining further **boosting downstream performance**.
- **Model ED > DFT ED**: Clarified that gains stem from inductive-bias alignment, not higher physical accuracy.
- **Other updates**: Added cross-domain background for non-experts and fixed typos.

All reviewers appreciated our rebuttal and follow-ups. All maintained **positive scores** with **high confidence (>=3)**, and explicitly recognized EDBench's significance.

In light of its novelty, technical depth, broad applicability, and extensive validation, we sincerely hope it may merit consideration for NeurIPS presentation. Thank you again for your thoughtful consideration.

**Best regards,**

**Authors**

---

### Decision · Program_Chairs · 2025-09-18

**Decision:**

Accept (poster)

**Comment:**

This paper introduces a new dataset called EDBench, which focuses on electron density (ED) for molecular modeling. It’s a large and well-curated dataset that fills an important gap in the field, as most existing benchmarks focus on simpler molecular properties. The reviewers appreciated the scale and quality of the data, the variety of tasks included, and the effort the authors made to ensure reproducibility. While there were initial concerns about the choice of computational methods and the models used for benchmarking, the authors responded thoroughly with additional experiments and clarifications that addressed these points well.

After the rebuttal, all reviewers either kept or raised their scores, showing strong support for the paper. They noted that the authors improved the clarity of the writing, added more relevant baselines, and extended the benchmark to cover periodic systems. Overall, the paper is technically solid and clearly presented.